# SPEECH LANGUAGE MODELS LACK IMPORTANT BRAIN-RELEVANT SEMANTICS

## ABSTRACT

Despite known differences between reading and listening in the brain, recent work has shown that text-based language models predict both text-evoked and speech-evoked brain activity to an impressive degree. This poses the question of what types of information language models capture that is correlated with features truly predict in the brain. We investigate this question via a direct approach, in which we eliminate information related to specific low-level stimulus features (textual, speech, and visual) in the language model representations, and observe how this intervention affects the alignment with fMRI brain recordings acquired while participants read versus listened to the same naturalistic stories. We further contrast our findings with speech-based language models, which would be expected to predict speech-evoked brain activity better, provided they model language processing in the brain well. Using our direct approach, we find that both text-based and speech-based language models align well with early sensory areas due to shared low-level features. Text-based models continue to align well with later language regions even after removing these features, while, surprisingly, speech-based models lose most of their alignment. These findings suggest that speech-based models can be further improved to better reflect brain-like language processing.

## 1 INTRODUCTION

An explosion of recent work that investigates the alignment between the human brain and language models shows that text-based language models (e.g. GPT*, BERT, etc.) predict both text-evoked and speech-evoked brain activity to an impressive degree (text: (Toneva & Wehbe, 2019; Schrimpf et al., 2021; Goldstein et al., 2022; Aw & Toneva, 2022; Oota et al., 2022; Lamarre et al., 2022; Chen et al., 2023); speech: (Jain & Huth, 2018; Caucheteux & King, 2020; Antonello et al., 2021; Vaidya et al., 2022; Millet et al., 2022; Tuckute et al., 2022; Oota et al., 2023a;b;c; Chen et al., 2023). This observation holds across late language regions, which are thought to process both text- and speech-evoked language (Deniz et al., 2019), but also more surprisingly across early sensory cortices, which are shown to be modality-specific (Deniz et al., 2019; Chen et al., 2023). Since text-based language models are trained on written text (Devlin et al., 2019; Radford et al., 2019; Chung et al., 2022), their impressive performance at predicting the activity (also referred to as alignment) in early auditory cortices is puzzling. This raises the question of what types of information underlie the brain alignment of language models observed across brain regions.

In this work, we investigate this question via a direct approach (see Fig. 1 for a schematic). For a number of low-level textual, speech, and visual features, we analyze how the alignment between brain recordings and language model representations is affected by the elimination of information related to these features. We further contrast our findings with speech-based language models, which would be expected to predict speech-evoked brain activity better, provided they model language processing in the brain well. For this purpose, we present a systematic study of the brain alignment across two popular fMRI datasets of naturalistic stories (1-reading, 1-listening) and different natural language processing models (text vs. speech). We focus on three popular text-based language models (BERT (Devlin et al., 2019), GPT2 (Radford et al., 2019), and FLAN-T5 (Chung et al., 2022)) and two speech-based models (Wav2vec2.0 (Baevski et al., 2020) and Whisper (Radford et al., 2022))–which have been studied extensively in the NLP-brain alignment literature (Toneva & Wehbe, 2019; Aw & Toneva, 2022; Merlin & Toneva, 2022; Oota et al., 2022; 2023a; Millet et al., 2022; Vaidya et al., 2022). The fMRI recordings are openly available (Deniz et al., 2019) and

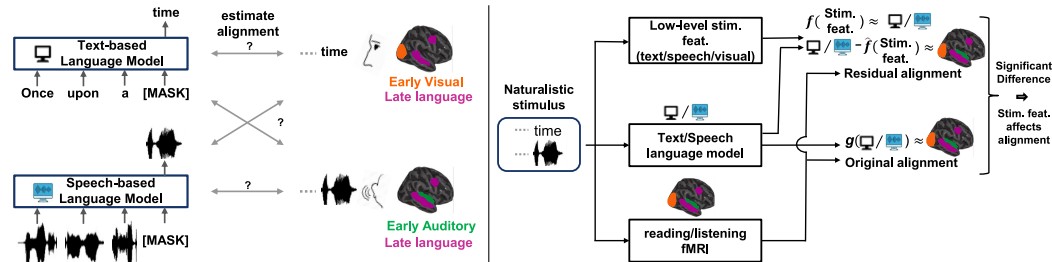

Figure 1: A direct approach to test the effect of low-level stimulus features on the alignment between different types of language models and brain recordings (reading vs. listening).

correspond to 6 participants reading and listening to the same naturalistic stories. This dataset is the only one of its kind, in which the participants are observing the same naturalistic stimuli but in two different modalities. It is important that the datasets we analyze have the same stimuli but are presented in two different modalities, because if the stimuli differ between modalities, any observed difference between modalities may be due to the difference in the presented stimuli and not to the difference in sensory modalities. We test how eliminating a range of low-level textual (number of letters, number of words, word length, etc.), speech (number of phonemes, Fbank, MFCC, Mel, articulation, phonological features, etc.), and visual features (motion energy) in artificial model representations affects alignment with brain responses.

Using our direct approach, we find that both types of language models predict brain responses well both during reading and listening in corresponding early sensory regions in large parts due to low-level stimulus features that are correlated between text and speech (e.g. number of letters and phonemes). In contrast, in later language regions, we find that much of the ability of text-based language models to predict brain responses is retained during both listening and reading even after removing low-level stimulus features, suggesting that text-based language models capture important brain-relevant semantics. However, surprisingly, we find that speech-based language models do not predict additional substantial variance in the late language regions, once low-level stimulus features are removed from the model representations.

Our main contributions can be summarized as follows: (i) We employ a direct approach to study the fine-grained aspects of information (i.e., low-level textual, speech, and visual features) in different types of neural language processing models and brains; (ii) We show that both text- and speech-based language models predict brain activity well both during reading and listening. During listening, speech-based models outperform text-based models in the auditory cortex. During reading, both text- and speech-based models have similar brain alignment with the early visual regions. However, the alignment with the late language regions is significantly better for text-based vs. speech-based models both during reading and listening; (iii) Our direct residual approach reveals that low-level stimulus features are the primary contributors underlying the alignment with early visual and auditory regions for both types of models. Specifically, diphones and number of letters are the two features that explain the most variance in brain responses for text-based models, while phonological features explain the most variance for speech-based models. We will make all code available upon publication so that other researchers can reproduce and build on our methodology and findings.

## 2 RELATED WORK

Our work is most closely related to that of (Toneva et al., 2022), who proposes the direct residual approach to study the supra-word meaning of language by removing the contribution of individual words to brain alignment. More recent work uses the same residual approach to investigate the effect of syntactic and semantic properties on brain alignment across layers of a text-based language model (Oota et al., 2023a). We build on this approach to investigate the contribution of low-level stimulus features to the brain alignment of both text-based and speech-based language models.

Our work also relates to a growing literature that investigates the alignment between human brains and language models. A number of studies have used text-based language models to predict both text-evoked and speech-evoked brain activity to an impressive degree (Wehbe et al., 2014; Jain & Huth, 2018; Toneva & Wehbe, 2019; Schwartz et al., 2019; Caucheteux & King, 2020; Jat et al.,

2020; Schrimpf et al., 2021; Goldstein et al., 2022; Antonello et al., 2021; Oota et al., 2022; Merlin & Toneva, 2022; Aw & Toneva, 2022; Oota et al., 2023a; Lamarre et al., 2022; Chen et al., 2023). Similarly, the recent advancements in Transformer-based models for speech (Chung et al., 2020; Baevski et al., 2020; Hsu et al., 2021) have motivated neuroscience researchers to test their brain alignment (Millet et al., 2022; Vaidya et al., 2022; Tuckute et al., 2022; Oota et al., 2023b;c; Chen et al., 2023) for speech-evoked brain activity. Our approach is complementary to these previous works and can be used to further understand what types of information underlie the brain alignment of language models, particularly across different brain regions. As demonstrated by (Deniz et al., 2019), early sensory cortices appear to be more modality-specific, and our approach can shed light on how these modality-specific aspects relate to brain alignment.

## 3 DATASETS AND MODELS

### 3.1 BRAIN DATASETS

We analyzed two fMRI datasets which were recorded while the same six participants listened to and read the same narrative stories selected from the Moth Radio Hour. These datasets were made publicly available by Deniz et al. (2019) and contain data for 6 participants. The brain responses of each participant contain 3737 samples (TRs) for training and 291 samples for testing. These datasets were selected because they are the only large naturalistic datasets for which the same stimulus was presented both auditorily and visually. We use a multi-modal parcellation of the human cerebral cortex (Glasser Atlas; 180 regions of interest (ROIs) in each hemisphere) (Glasser et al., 2016). This includes three early sensory and four language-relevant ROIs in the human brain with the following subdivisions: (1) early visual (V1, V2); (2) visual word form area (VWFA: PH and TE2P); (3) early auditory area (A1, PBelt, MBelt, LBelt, RI, A4); and (4) late language regions encompassing broader language regions: angular gyrus (AG: PFm, PGs, PGi, TPOJ2, TPOJ3), lateral temporal cortex (LTC: STSda, STSva, STGa, TE1a, TE2a, TGv, TGd, A5, STSdp, STSvp, PSL, STV, TPOJ1), inferior frontal gyrus (IFG: 44, 45, IFJa, IFSp) and middle frontal gyrus (MFG: 55b) (Baker et al., 2018; Milton et al., 2021; Desai et al., 2022). The functionality of these ROIs is reported in the Appendix (see Table 1).

**Estimating dataset noise ceiling.** To account for the intrinsic noise in biological measurements and obtain a more informative estimate of a model's brain alignment, we follow previous work Schrimpf et al. (2021) to estimate the ceiling value for a model's performance for the reading and listening fMRI datasets. This is achieved by subsampling the fMRI datasets with 6 recorded participants. Specifically, we create all possible combinations of $s$ participants ($s \in [2,6]$), separately for reading and listening. For each subsample $s$, we estimate the amount of brain response in one target participant that can be predicted using only data from other participants, using a voxel-wise encoding model (see Sec. 4). Note that the estimated noise ceiling is based on the assumption of a perfect model, which may not always be the case in real-world scenarios. Nonetheless, this approach can put the model's performance in a useful perspective. We present the average estimated noise ceiling across voxels for the *naturalistic reading-listening fMRI* dataset in Appendix AFig. 7. We observe that the average estimated noise ceiling across voxels for the two modalities is not significantly different. However, as depicted in Fig 2, there are clear regional differences: Early visual areas have higher noise ceiling during the reading condition (red voxels), while many of the early auditory areas have a higher noise ceiling during the listening condition (blue voxels).

### 3.2 LANGUAGE MODELS

To investigate the reasons for brain alignment of language models during reading and listening, we extract activations from five popular pretrained Transformer models. Three of these models are "text-based": BERT-base, GPT2-small, and FLAN-T5; and two are "speech-based": Wav2Vec2.0-base and Whisper-base. Below we present more details for each model.

**Text-based language models.** To extract representations of the text stimulus, we use three popular pretrained Transformer language models from Huggingface (Wolf et al., 2020): (1) BERT (Devlin et al., 2019) uses only encoder blocks of standard Transformer-based architecture with 12 layers and 768-dimensional representations. (2) GPT-2 (Radford et al., 2019) uses only decoder blocks of

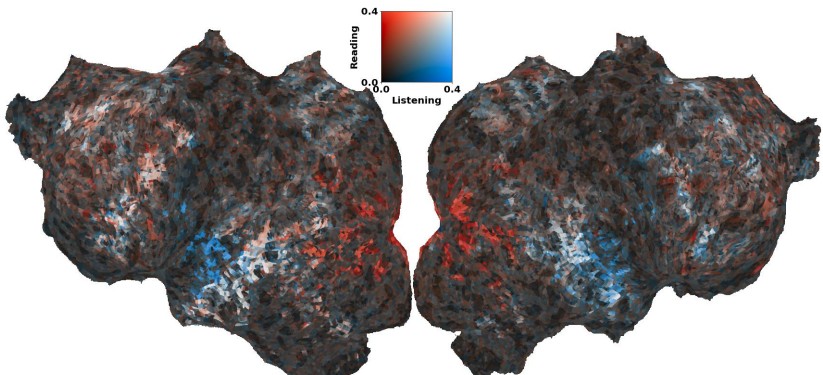

Figure 2: Contrast of estimated noise ceilings for a representative participant for the reading vs listening condition for one subject (subject-8). Blue voxels depict higher noise ceiling estimate in listening condition. Red voxels depict higher noise ceiling estimate in reading. Voxels that appear in white have similar noise ceilings during reading and listening, and are distributed across language regions. Noise ceilings for other participants are reported in Appendix A Figs. 8 and 9.

standard Transformer-based architecture with 12 layers and 768-dimensional representations. Here, the Transformer model was pretrained with next word prediction. (3) FLAN-T5 (encoder-decoder model) (Chung et al., 2022) uses text-text to Transformer model with 24 layers (12 layers encoder and 12 layers decoder) and 768-dimensional representations.

**Extracting text representations.** We follow previous work to extract the hidden-state representations from each layer of these language models, given a fixed-length input length (Toneva & Wehbe, 2019). To extract the stimulus features from these pretrained models, we constrained the tokenizer to use a maximum context of previous 20 words. Given the constrained context length, each word is successively input to the network with at most $C$ previous tokens. For instance, given a story of $M$ words and considering the context length of 20, while the third word's vector is computed by inputting the network with $(w_1, w_2, w_3)$, the last word's vectors $w_M$ is computed by inputting the network with $(w_{M-20}, \ldots, w_M)$. The pretrained Transformer model outputs token representations at different layers. We use the #tokens $\times$ 768 dimension vector obtained from the each hidden layer to obtain word-level representations from each pretrained Transformer language model. Since the rate of fMRI data acquisition (TR = 2.0045sec) was lower than the rate at which the text stimulus was presented to the subjects, several words fall under the same TR in a single acquisition. Hence, we match the stimulus acquisition rate to fMRI data recording by downsampling the stimulus features using a 3-lobed Lanczos filter. After downsampling, we obtain the chunk-embedding corresponding to each TR.

**Speech-based language models.** Similar to text-based language models, we use two popular pretrained Transformer speech-based models from Huggingface (Wolf et al., 2020): (1) Wav2Vec2.0 (encoder model) (Baevski et al., 2020) uses only encoder blocks of standard Transformer-based architecture with 12 layers and 768-dimensional representations. Here, the Transformer model was pretrained with contrastive loss as objective function. (2) Whisper (encoder-decoder model) (Radford et al., 2022) uses speech-to-speech Transformer model with 24 layers (12 layers encoder and 12 layers decoder) and 768-dimensional representations. Here, the Transformer model was pretrained and fine-tuned with multiple speech tasks.

**Extracting speech representations.** The input audio story is first segmented into clips corresponding to 2.0045 seconds, which matches the fMRI image rate. Each audio clip is input to the speech-based models one by one to obtain stimulus representations per clip. The representations are obtained from the activations of the pretrained speech model in intermediate layers. For all models, we used the checkpoints provided by the huggingface library (Wolf et al., 2020). Overall, each layer of speech-based models (Wav2Vec2.0 and Whisper) outputs 768 dimensional vector at each TR.

## 3.3 INTERPRETABLE STIMULUS FEATURES

To better understand the contribution of different stimulus features to the brain alignment of language models, we extract a range of low-level textual, speech, and visual features that have been

shown in previous work to relate to brain activity during listening and reading (see Sec. 2 for a summary of the related work).

**Low-level textual features.** (1) Number of Letters: The number of letters in each TR. (2) Number of Words: The number of words in each TR. (3) Word Length STD: The standard deviation of word length corresponding to each TR. These low-level textual features, which are already downsampled and aligned with each TR, have also been used in (Deniz et al., 2019; Huth et al., 2022).

**Low-level speech features.** We extract low-level speech features like filter banks (FBank), Mel Spectrogram, and MFCC from audio files using Self-Supervised Speech Pre-training and Representation Learning (S3PRL) toolkit[1], and phonological features using the DisVoice library[2]. We also use the articulation and power spectrum (PowSpec) feature vectors provided in (Deniz et al., 2019; LeBel et al., 2022). (1) Number of Phonemes: The number of phonemes in each TR. (2) Mono-Phones: The smallest unit of speech (e.g., /p/, /c/, /a/) distinguishing one word (or word element) from another. There are 39 phonemes in English, and we generate a 39-dimensional feature vector at each TR. (3) DiPhones: Diphones represent the adjacent pair of phones (e.g., [da], [aI], [If]) in an utterance. For each TR, we obtained a one-hot code encoding the presence or absence of all possible diphones (858). (4) TriPhones: A triphone is a sequence of three consecutive phonemes. (5) FBank: Filter banks divide the raw audio signal into multiple components (each one carrying a single frequency sub-band of the original signal) using a bandpass filter, results in a 26-dimensional vector. (6) Mel Spectrogram: Mel spectrogram features are computed by applying a Fourier transform on the raw audio signal to analyze a signal's frequency content and converting it to the mel-scale, yielding an 80-dimensional vector. (7) MFCC: MFCC features are Mel-frequency spectral coefficients obtained by taking the Discrete Cosine Transform (DCT) of the spectral envelope obtained from the logarithmic filter bank outputs. (8) PowSpec: The time-varying power spectrum features are provided in (Gong et al., 2023). They were obtained by estimating the power for each 2-s segment of the audio signal between 25 Hz and 15 kHz, in 33.5 Hz bands (number of frequency bands = 448). (9) Phonological: Phonological features are the smallest units of distinction between any two phonemes. We compute 108 phonological features consisting of (18 descriptors, e.g. vocalic, consonantal, back, etc,.)×(6 functionals: mean, std, skewness, kurtosis, max, min). (10) Articulation: We use phoneme articulations as a mid-level speech feature. These were derived by mapping hand-labeled phonemes onto 22 articulatory features.

**Low-level visual features.** (1) Motion energy features: We use motion energy features as low-level visual features. These features were generated through the utilization of a spatiotemporal Gabor pyramid. This pyramid was employed to extract low-level visual characteristics from the sequence of word frames employed in the reading experiment (Adelson & Bergen, 1985). The resulting motion energy features were (39 parameters), as provided in Deniz et al. (2019).

## 4 METHODOLOGY

Our direct approach to investigate the reasons for brain alignment of language models involves three main steps (see Fig. 1): (1) removal of interpretable stimulus features from the language model representations; (2) estimating the brain alignment of the language model representations before and after removal of a particular feature; (3) a significance test to conclude whether the difference in estimated brain alignment before and after is significant.

**Removal of low-level features from language model representations.** To remove low-level features from language model representations, we rely on a simple method proposed previously by Toneva et al. (2022); Oota et al. (2023a), in which the linear contribution of the feature to the language model activations is removed via ridge regression. In our setting, we remove the linear contribution of a low-level feature by training a ridge regression, in which the low-level feature vector is considered as input and the neural word/speech representations are the target. We compute the residuals by subtracting the predicted feature representations from the actual features resulting in the (linear) removal of low-level feature vector from pretrained features. Because the brain prediction method is also a linear function (see next paragraph), this linear removal limits the contribution of low-level features to the eventual brain alignment.

---

[1] https://github.com/s3prl/s3prl
[2] https://github.com/jcvasquezc/DisVoice

Specifically, given an input feature vector $\mathbf{L}_i$ with dimension $\mathbf{N} \times \mathbf{d}$ for low-level feature $i$, and target neural model representations $\mathbf{W} \in \mathbb{R}^{\mathbf{N} \times \mathbf{D}}$, where $\mathbf{N}$ denotes the number of TRs, d and D denote the dimensionality of low-level and neural model representations, respectively. Overall, the ridge regression objective function is $f(\mathbf{L}_i) = \min_{\theta_i} \|\mathbf{W} - \mathbf{L}_i \theta_i\|_F^2 + \lambda \|\theta_i\|_F^2$ where $\theta_i$ denotes the learned weight coefficient for embedding dimension $\mathbf{D}$ for the input feature $i$, $\|.\|_F^2$ denotes the Frobenius norm, and $\lambda > 0$ is a tunable hyper-parameter representing the regularization weight for each feature dimension. Using the learned weight coefficients, we compute the residuals as follows: $r(\mathbf{L}_i) = \mathbf{W} - \mathbf{L}_i \theta_i$.

**TR Alignment.** To account for the slowness of the hemodynamic response, we model the hemodynamic response function using finite response filter (FIR) per voxel and for each subject separately with 6 temporal delays corresponding to 12 seconds.

**Voxel-wise encoding model.** We estimate the brain alignment of a language model before and after the removal of a stimulus property via training standard voxel-wise encoding models Deniz et al. (2019); Toneva & Wehbe (2019). Specifically, for each voxel and participant, we train a ridge regression model to predict the fMRI recording associated with this voxel as a function of the stimulus representations obtained from both text and speech-based models (before and after the removal of stimulus features). In particular, we use layerwise pretrained representations from both text and speech-based models as well as residuals by removing each basic low-level feature and using them in a voxelwise encoding model to predict brain responses. If the removal of a particular stimulus property from the language model representation leads to a significant drop in brain alignment, then we conclude that this stimulus property is important for the brain alignment of the language model. In this paper, we train fMRI encoding models using Banded ridge regression (Tikhonov et al., 1977). Before the regression, we first z-scored each feature channel separately for training and testing. This was done to match the features to the fMRI responses, which were also z-scored for training and testing. Formally, at the time step (t), we encode the stimuli as $X_t \in \mathbb{R}^{N \times D}$ and brain region voxels $Y_t \in \mathbb{R}^{N \times V}$, where $N$ denotes the number of training examples, $D$ denotes the dimension of the concatenation of delayed 6 TRs, and $V$ denotes the number of voxels. To find the optimal regularization parameter for each feature space, we use a range of regularization parameters that is explored using cross-validation. The main goal of each fMRI encoding model is to predict brain responses associated with each brain voxel given a stimulus.

**Normalized predictivity.** The final measure of a model's performance is obtained by calculating the Pearson's correlation between the model's predictions and neural recordings. This correlation is then divided by the estimated ceiling and averaged across voxels, regions and participants, resulting in a standardized measure of performance referred to as normalized predictivity.

**Implementation details for reproducibility.** All experiments were conducted on a machine with 1 NVIDIA GEFORCE-GTX GPU with 16GB GPU RAM. We used banded ridge-regression with the following parameters: MSE loss function, and L2-decay ($\lambda$) varied from $10^1$ to $10^3$; the best $\lambda$ was chosen by using the validation data in each cross-validation fold.

## 5 RESULTS

We calculate the normalized brain predictivity independently for text- and speech-based models, averaging the results within each model category separately. Likewise, we compute the normalized predictivity for each specific low-level stimulus feature type (e.g., textual), considering only the corresponding low-level stimulus features (e.g., number of letters). Except for the qualitative analysis, all our results presented in subsections 5.1 and 5.2 are averaged within types of models and types of low-level stimulus feature categories.

### 5.1 TEXT VS. SPEECH MODEL ALIGNMENT DURING READING VS. LISTENING

We assess the degree to which each type of model aligns with different regions of interest (ROI) across the brain. Specifically, we consider early sensory regions (early visual and early auditory) in addition to the visual word form area (VWFA) and late language regions.

In Fig. 3, we present the brain alignment of each model normalized by the noise ceiling for both reading and listening. We show the average normalized brain predictivity computed across partic-

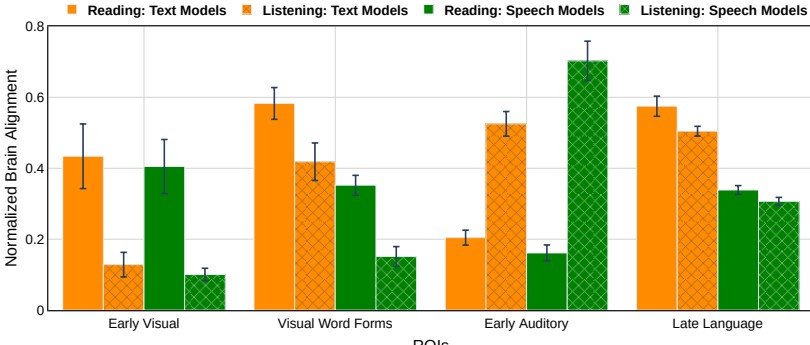

Figure 3: ROI-based average normalized brain predictivity was computed across participants, models, layers and voxels. In the case of brain reading, we represent the data using full-color bars, while for listening, we use bars with patterns. Likewise, we refer to the Orange-text and the Green-speech language models.

ipants, models, layers, and voxels for both text- and speech-based language models. Specifically, we report the average normalized predictivity across text-based models (BERT, GPT2, and FLAN-T5) and speech-based models (Wav2Vec2.0 and Whisper). To test whether the difference in brain alignment is statistically significant between the two types of models, we perform the *Wilcoxon signed-rank* test.

We make the following observations from Fig. 3: (1) During reading, both text- and speech-based language models predict early visual areas similarly well. This result is quite surprising, as one might anticipate text-based models to better predict these areas than speech-based models. However, text-based models outperform speech-based models in VWFA. This implies that text-based models capture brain-relevant information related to processing of visual word forms, as VWFA has a specific computational role in decoding written forms of words and is considered a crucial node of the brain's language network (Dehaene & Cohen, 2011; McCandliss et al., 2003). (2) During listening, speech-based models outperform text-based models in the early auditory cortex. Nevertheless, text-based language models still predict brain activity to an impressive degree. This raises questions about what types of information text-based language models capture that is correlated with features relevant to early auditory processing. (3) In contrast, in late language regions, text-based models significantly outperform speech-based models during both reading and listening. However, the differences between text- and speech-based model alignment remain unclear, which we explore further via our direct residual approach.

## 5.2 DISSECTING BRAIN ALIGNMENT

While the previous analyses demonstrate that both text and speech-based language models predict brain activity to an impressive degree in the early sensory processing regions to late language regions during reading and listening, a major goal of the current study is to identify the specific types of information these language models capture in brain responses. To achieve this, we remove information related to specific low-level stimulus features (textual, speech, and visual) in the language model representations, and observe how this perturbation affects the alignment with fMRI brain recordings acquired while participants read versus listened to the same naturalistic stories. We present the results of these analyses for the early visual areas and VWFA during reading, the early auditory area during listening, and the late language regions during both reading and listening.

**Why do text-based language models predict speech-evoked brain activity in early auditory cortices?** In Fig. 4(a), we report the normalized brain predictivity results during listening in the early auditory cortex for both text- and speech-based language models, along with their residual performance after eliminating low-level stimulus features. We make the following observations: (1) Removal of low-level textual features results in a similar performance drop for both types of models. (2) Removal of low-level speech features results in a larger performance drop for speech-based compared to text-based language models (more than 40% of the original performance). These findings indicate that speech-based language models outperform text-based language models because they better leverage low-level speech features such as fbank, MFCC, Mel spectrogram, and PowSpec. However, the presence of correlated information related to low-level textual (e.g., number of letters)

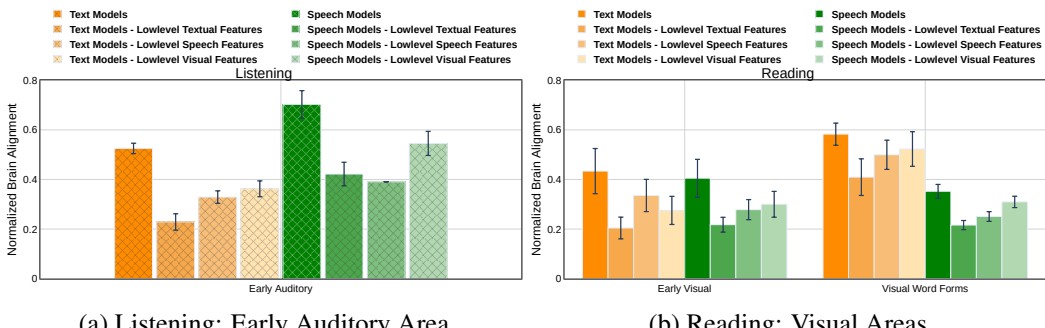

(a) Listening: Early Auditory Area        (b) Reading: Visual Areas

Figure 4: *(a)* Listening condition in the Early Auditory Area: Average normalized brain predictivity was computed across participants for text- and speech-based models, across layers and voxels in the early auditory cortex. *(b)* Reading condition in the Visual Areas: Average normalized brain predictivity was computed over the average of participants for text- and speech-based models, across layers and voxels in the early visual area and VWFA.

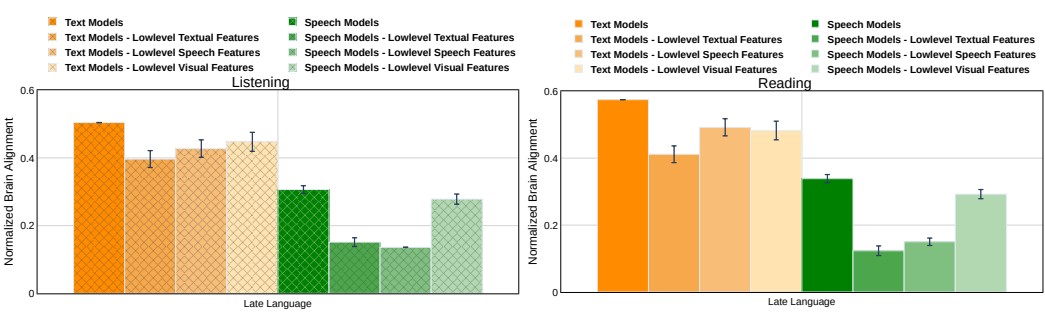

(a) Listening: Late Language Regions        (b) Reading: Late Language Regions

Figure 5: *(a)* Listening condition in the Late Language: Average normalized brain predictivity was computed across participants for text- and speech-based models, across layers and voxels. *(b)* Reading condition in the Late Language: Average normalized brain predictivity was computed over the average of participants for text- and speech-based models, across layers and voxels.

and speech (e.g., number of phonemes) features explains a large portion of the brain alignment for both types of models. Additionally, we find that the removal of low-level visual features (motion energy) has a much smaller effect on the alignment of both types of models. This is possibly because the features in language models are not correlated with visual stimulus features, such as edges. We present the residual performance results of individual low-level stimulus features for both text- and speech-based language models in the Appendix (see Fig. 11).

**Why do both types of models exhibit similar degree of brain alignment in early visual cortices?** We make the following observations from Fig. 4(b): (1) Similar to the listening condition in the early auditory areas, the removal of low-level textual features from both types of models leads to a significant drop in brain alignment. This indicates that the performance of both types of models in early visual cortices is largely due to low-level textual features that are correlated with low-level speech features (see Fig. 12 in the Appendix). Additionally, the removal of low-level visual features has a much smaller effect for both types of models in both early visual cortices and VWFA, presumably because these models are not designed to process low-level visual information. (2) Text-based models outperform speech-based models in VWFA, as this region is mainly associated with processing visual word forms and speech-based models are not equipped to handle this type of information. We present the residual performance results of individual low-level stimulus features for both text- and speech-based language models in the Appendix (see Fig. 12).

**Are there any possible differences between text- and speech-based models in late language regions?** In Fig. 5(a) and (b), we report the normalized brain predictivity of both text- and speech-based language models and their residual performance after removal of low-level stimulus features for reading and listening in the late language regions. Text-based models explain a large amount

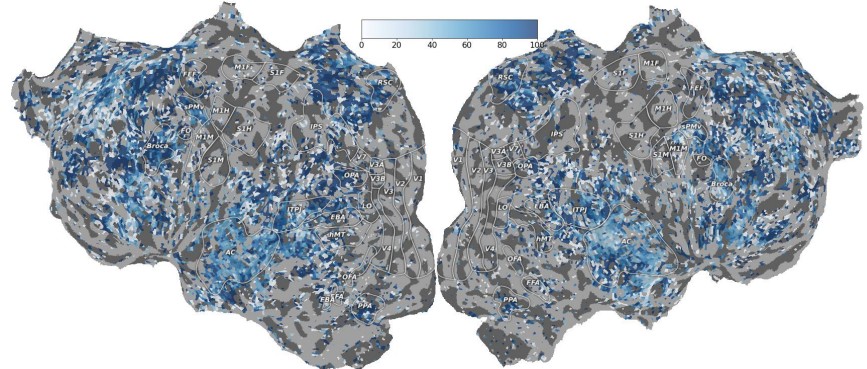

Figure 6: Listening: Percentage of decrease in alignment for each voxel after the removal of Phonological features from Wav2Vec2.0. Percentage decrease in alignment for each voxel in one subject (subject-8) are projected onto the subject's flattened cortical surface. Voxels appear white if phonological features do not explain any shared information of Wav2Vec2.0, and blue if all the information predicted by Wav2Vec2.0 is similar to the information captured using phonological features. Voxels appear in light blue indicates that parts of AC explained (40-60% drop) by phonological features and there are additional features in Wav2Vec2.0 that are relevant for early auditory processing.

of variance in these regions, even after removing low-level stimulus features. In contrast, residual performance of speech-based models goes down to around 10-15%. This implies that the alignment of speech-based models with late language regions is almost entirely due to low-level stimulus features, and not brain-relevant semantics.

**Qualitative analysis**. To determine the low-level stimulus feature that has the highest impact on brain predictivity during reading and listening, we conducted a fine-grained analysis where we measured the drop in brain predictivity for each voxel after removing low-level stimulus features from text- and speech-based models. We found that the "Phonological" feature had the highest impact during listening for the speech-based model Wav2Vec2.0 and "Number of Letters" had the highest impact during reading for the text-based model BERT. Figs. 6 and 14 (see Appendix) display the percentage decrease in brain predictivity for listening (Wav2Vec2.0 with Phonological) and reading (BERT with Number of Letters), respectively. Removing "Number of Letters" leads to a significant drop (80-100%) in the early visual areas, but only a slight drop (0-20%) in the late language areas during reading. Fig. 15 (see Appendix) displays the percentage decrease in predictivity when the low-level speech feature "DiPhones" is removed from BERT representations during reading. Since many common short words are composed of diphones (Gong et al., 2023), removing this feature from BERT significantly decreases predictivity (20-40%) even in late language regions. Removing Phonological features from Wav2Vec2.0 leads to a substantial drop (80-100%) in performance in the late language regions. This indicates that there is little, if any, brain-relevant information in speech-based models beyond low-level speech features in late language regions.

## 6 DISCUSSION AND CONCLUSION

We propose a direct approach to evaluate what types of information language models capture that is correlated with features truly predict in brain responses. This is achieved by eliminating the information related to specific low-level stimulus features (textual, speech, and visual) and observing how this intervention affects the alignment with fMRI brain recordings acquired while participants read versus listened to the same naturalistic stories. We show that both types of language models predict brain responses well both during reading and listening in corresponding early sensory areas in large parts due to low-level stimulus features that are correlated between text and speech (e.g., number of letters and phonemes). We found that text-based models predict fMRI recordings significantly better than speech-based models, irrespective of stimulus modality. These findings suggest that speech-based models can be further improved to better reflect brain-like language processing.

Our current study primarily focuses on the effect of low-level stimulus features on the alignment between language models and the human brain. However, the human brain also processes high-level semantic features (e.g., discourse-level or emotion-related). Future work can also examine how such high-level semantic features affect this alignment. Since semantic comprehension encompasses elements like metaphors, humor, sarcasm, discourse, emotion, and narrative information.

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

# *Appendix for: Speech language models lack important brain-relevant semantics*

## A ESTIMATED NOISE CEILING

We present the average estimated noise ceiling across voxels for the *naturalistic reading-listening fMRI* dataset in Fig. 7. We observe that the average estimated noise ceiling across voxels for the two modalities is not significantly different. However, as depicted in Figs. 8 and 9, there are clear regional differences across all the participants: Early visual regions have higher noise ceiling during the reading condition (red voxels), while many of the early auditory regions have a higher noise ceiling during the listening condition (blue voxels).

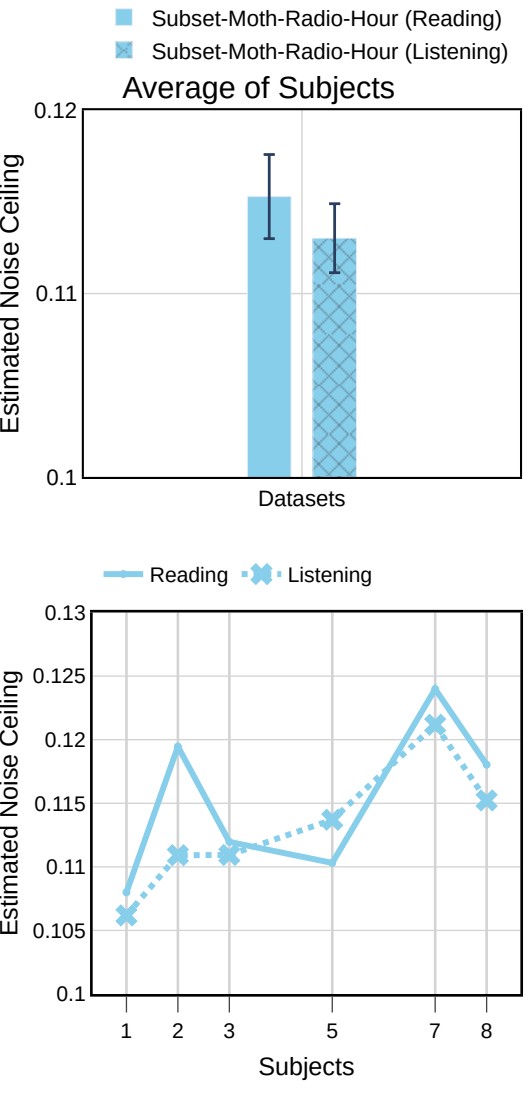

Figure 7: The estimated noise ceiling was computed across all subjects for the Subset-Moth-Radio-Hour naturalistic reading-listening fMRI dataset. The average noise ceiling is shown across predicted voxels where each voxel ceiling value is $> 0.05$.

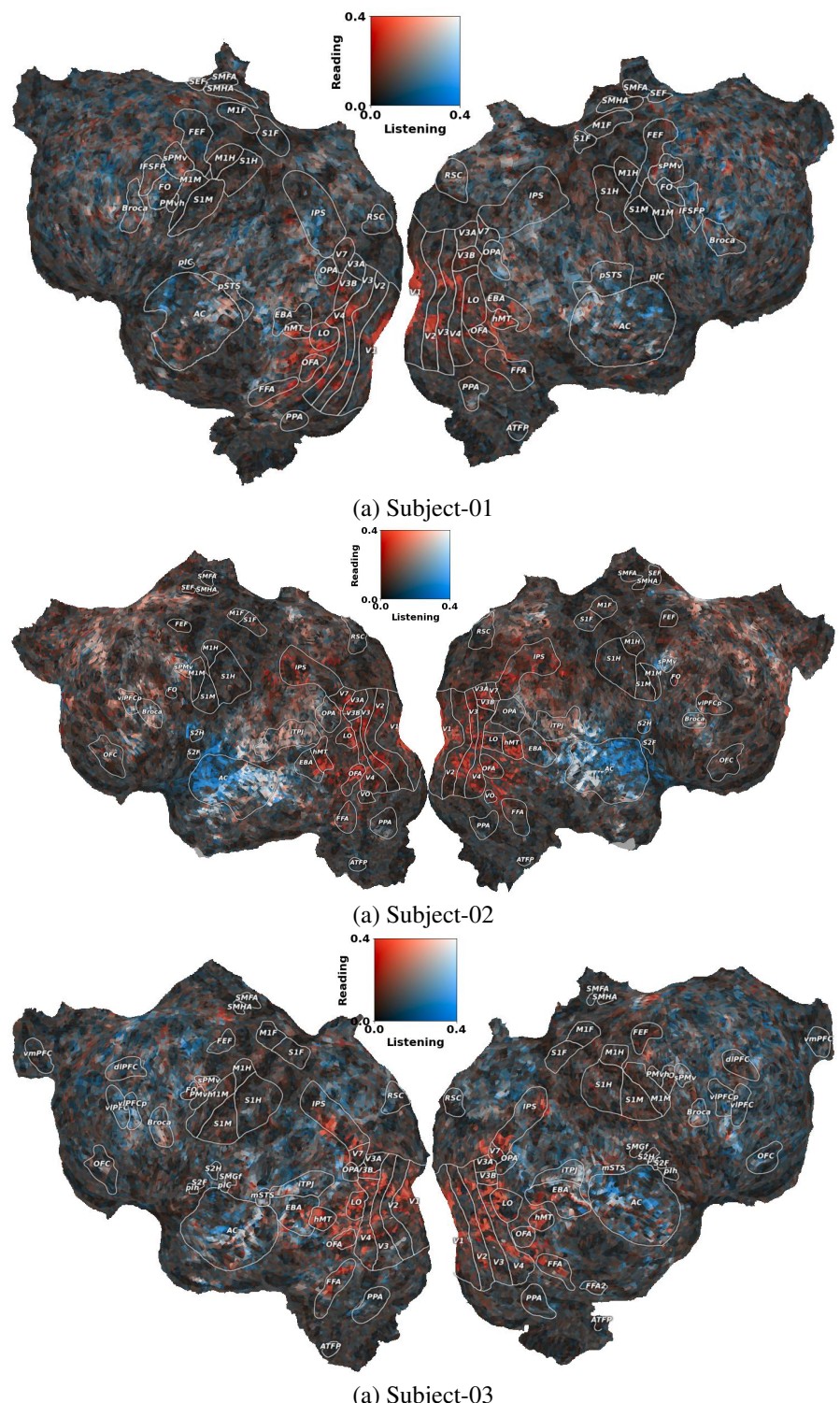

(a) Subject-01

(a) Subject-02

(a) Subject-03

Figure 8: Contrast of estimated noise ceilings for the remaining participants for the reading vs listening condition. BLUE-AC (Auditory Cortex) voxels have a higher noise ceiling in listening, and Red-VC (Visual Cortex) voxels have a higher noise ceiling in reading. Voxels that appear in white have similar noise ceilings across conditions, and are distributed across language regions.

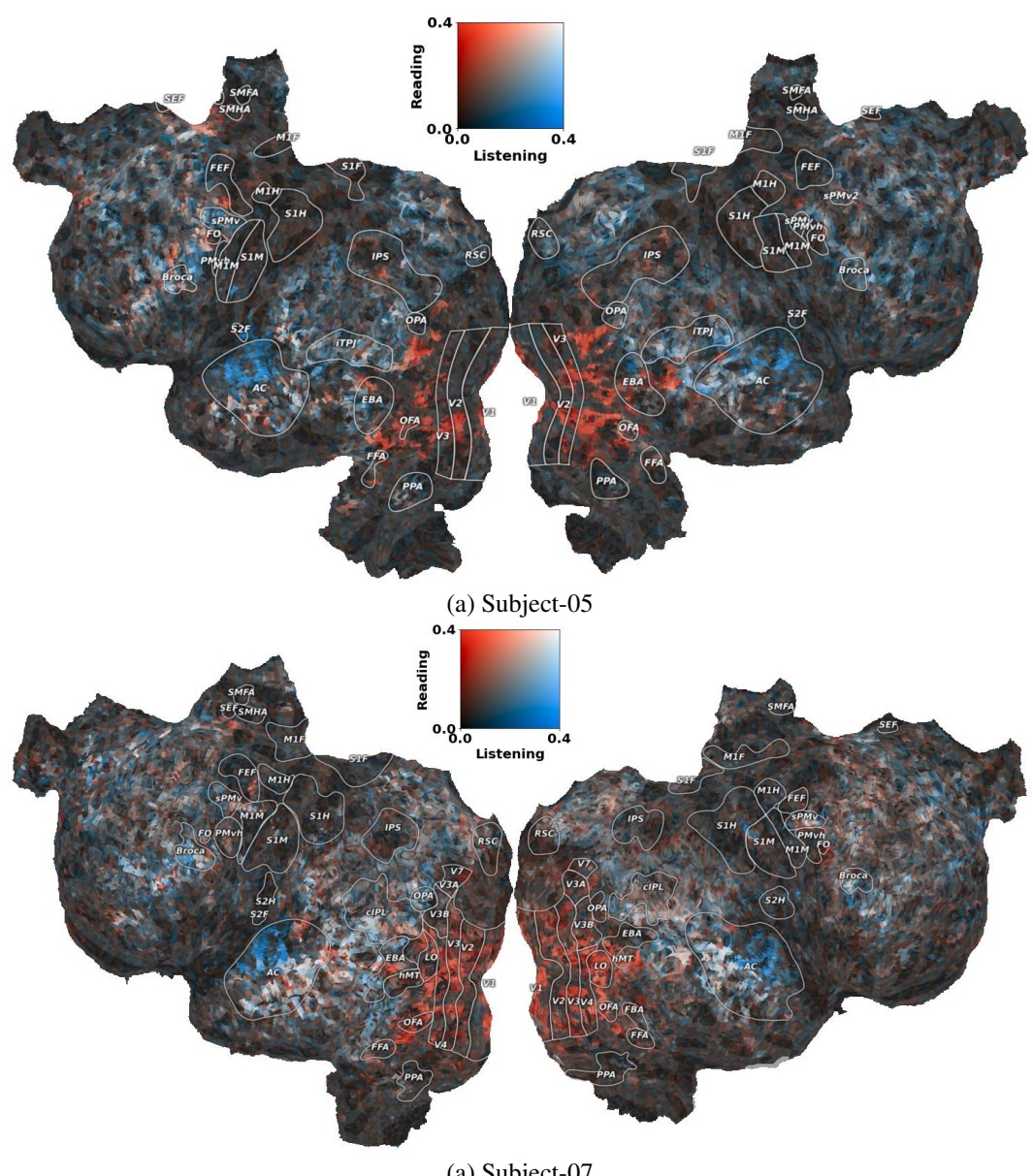

(a) Subject-05

(a) Subject-07

Figure 9: Contrast of estimated noise ceilings for the remaining participants for the reading vs listening condition. BLUE-AC (Auditory Cortex) voxels have a higher noise ceiling in listening, and Red-VC (Visual Cortex) voxels have a higher noise ceiling in reading. Voxels that appear in white have similar noise ceilings across conditions, and are distributed across language regions.

# B WHOLE BRAIN ANALYSIS: TEXT VS. SPEECH MODEL ALIGNMENT DURING READING VS. LISTENING

In Fig. 10, we report the whole brain alignment of each model normalized by the noise ceiling for the naturalistic reading and listening datasets. We show the average normalized brain alignment across subjects, layers, and voxels. Note that we are only averaging across voxels which have a statistically significant brain alignment. We perform the *Wilcoxon signed-rank* test to test whether the differences between text- and speech-based language models are statistically significant. We found that all text-based models predict brain responses significantly better than all speech-based models in both modalities. Across text-based models, the whole brain alignment gradually diminishes when going from BERT to GPT-2 to FLAN-T5 both during reading and listening. Across speech-based models, stimulus modality had a significant effect. While Wav2Vec2.0 is the better performer during reading,

Whisper aligns better with the whole brain during listening. The fact that Whisper is trained on a larger amount of speech data could be the reason underlying its better alignment during listening.

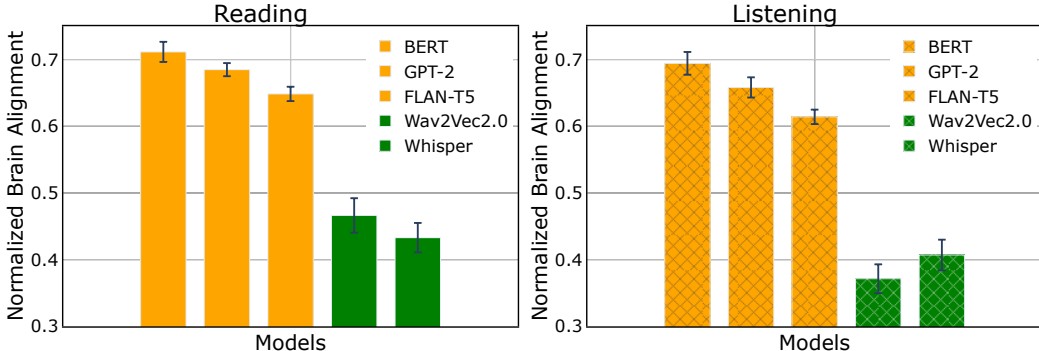

Figure 10: Average normalized brain predictivity was computed over the average of subjects for each model, across layers (3 text-based language models, 2 modalities (reading and listening), and 2 speech-based models).

## C  DISSECTING BRAIN ALIGNMENT

Our major goal of the current study is to identify the specific types of information these language models capture in brain responses. To achieve this, we remove information related to specific low-level stimulus features (textual, speech, and visual) in the language model representations, and observe how this perturbation affects the alignment with fMRI brain recordings acquired while participants read versus listened to the same naturalistic stories. In subsections 5.1 and 5.2 (see in the main paper), all our results presented are averaged within types of models and types of low-level stimulus feature categories. Here, we report the residual performance results of individual low-level stimulus features for both text- and speech-based language models, as shown in Figs. 11 and 12.

### C.1  WHY DO TEXT-BASED LANGUAGE MODELS PREDICT SPEECH-EVOKED BRAIN ACTIVITY IN EARLY AUDITORY CORTICES?

In Fig. 11, we report the normalized brain alignment results during listening in the early auditory cortex for both text- and speech-based language models, along with their residual performance after eliminating low-level stimulus features.

**Removal of low-level textual features** We make the following observations from Fig. 11 (a): (1) Removal of number of letters feature results in a larger performance drop (more than 30% of the original performance) for both text- and speech-based language models. (2) Similarly removal of number of words feature also affect more than 25% drop indicate that low-level textual features are captured in both text and speech-based language models.

**Removal of low-level speech features** We make the following observations from Fig. 11 (b): (1) Removal of phonological features results in a larger performance drop (more than 50% of the original performance) for speech-based language models than text-based models (30% drop of the original performance). (2) Additionally, the removal of low-level speech features such as Mel spectrogram, MFCC and DiPhones leads to major performance drop (more than 40%) for speech-based language models. (3) In contrast, the removal of remaining low-level speech features, including FBANK, PowSpec and Articulation, has less shared information with speech-based language models and results in a minor performance drop (i.e. less than 20%). These findings indicate that speech-based language models outperform text-based language models because they better leverage low-level speech features such as MFCC, Mel spectrogram, and Phonological. Overall, phonological features are the largest contributors for for both text and speech-based language models. Specifically, the presence of correlated information in phonological features related to low-level textual (e.g., number of letters) and speech (e.g., number of phonemes) features explains a large portion of the brain alignment for both types of models.

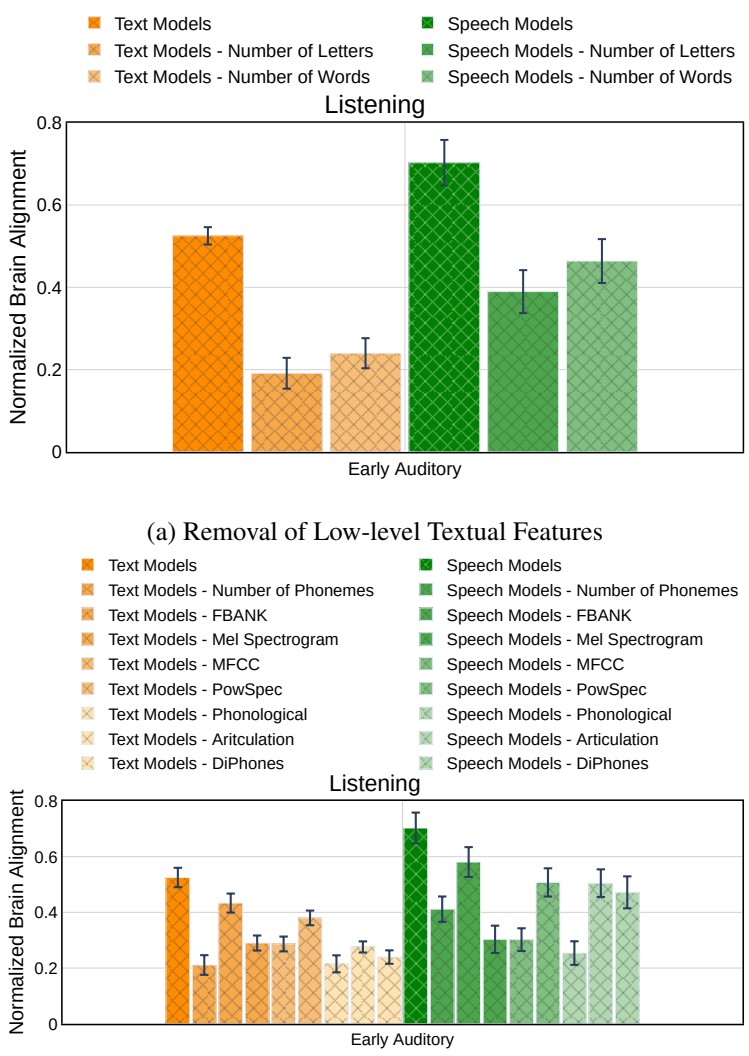

(a) Removal of Low-level Textual Features

(b) Removal of Low-level Speech Features

Figure 11: Brain Listening: *(a)* Removal of Low-level textual features, Average normalized brain predictivity was computed over the average of participants for text and speech-based models, across layers for each low-level textual property. *(b)* Removal of low-level speech features: Average normalized brain predictivity was computed across participants for text and speech-based models, across layers for each low-level speech property.

## C.2 WHY DO BOTH TYPES OF MODELS EXHIBIT SIMILAR DEGREE OF BRAIN ALIGNMENT IN EARLY VISUAL CORTICES?

In Fig. 12, we report the normalized brain alignment results during reading in the early visual cortex for both text- and speech-based language models, along with their residual performance after eliminating low-level stimulus features.

**Removal of low-level textual features** We make the following observations from Fig. 12 (a): (1) Similar to the listening condition in the early auditory regions, the removal of number of letters feature from both types of models leads to a significant drop in brain alignment in the early visual region. (2) Furthermore, the removal of number of words feature also leads to a drop of more than 20% in the early visual region indicate that low-level textual features are captured in both text and speech-based language models. This indicates that the performance of both types of models in early visual cortices is largely due to the number of letters feature followed by the number of words.

**Removal of low-level speech features** We make the following observations from Fig. 12 (b): (1) In the early visual region, removal of phonological features results in a larger performance drop (more than 35% of the original performance) for speech-based language models than text-based models (20% drop of the original performance). (2) However, the removal of remaining low-level speech features has less shared information with text-based language models and results in a minor performance drop (i.e. less than 10%). (3) In the visual word form area, the removal of all low-level speech features has no affect on brain alignment for text-based language models, while the removal of phonological features from speech-based models results in alignment dropping to zero. Overall, phonological features are the largest contributors for speech-based language models, both in early visual and visual word form areas.

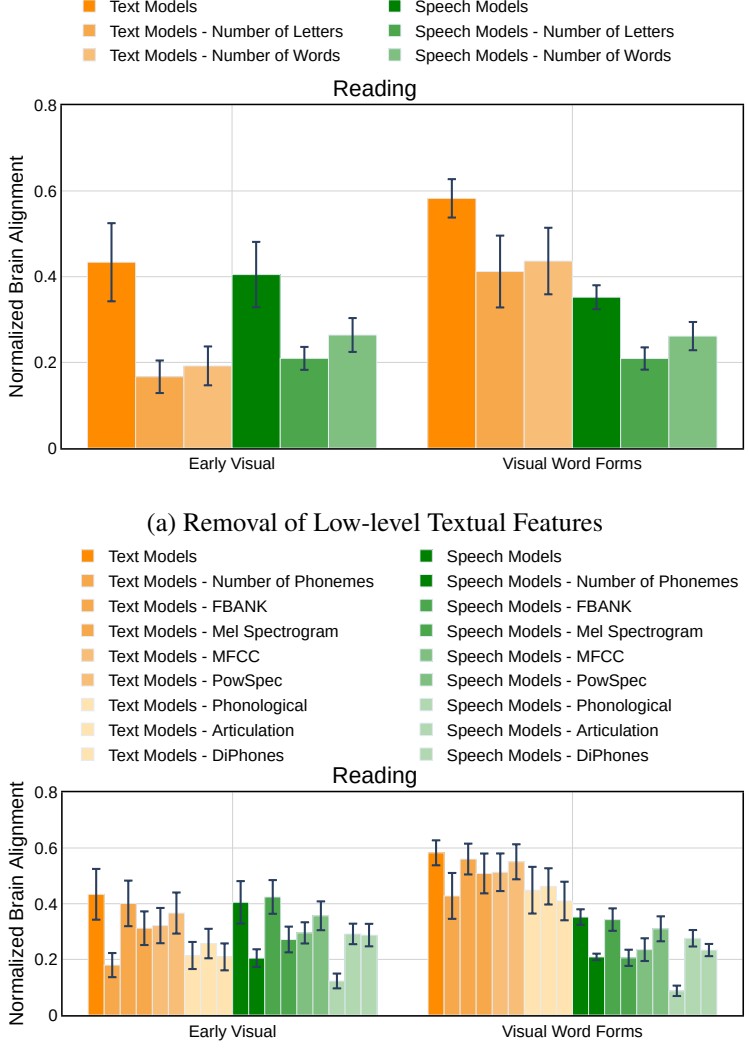

(a) Removal of Low-level Textual Features

(b) Removal of Low-level Speech Features

Figure 12: Brain Reading: *(a)* low-level textual features, Average normalized brain predictivity was computed over the average of participants for text and speech-based models, across layers for each low-level textual property. *(b)* low-level speech features: Average normalized brain predictivity was computed across participants for text and speech-based models, across layers for each low-level speech property.

**Are there any differences between text- and speech-based models in late language regions?**

**Removal of low-level textual features** In both reading and listening conditions, we make the following observations from Fig. 13 (a) and (c): (1) Text-based models explain a large amount of

variance in late regions, even after removing low-level textual features. (2) In contrast, residual performance of speech-based models goes down to approximately 10-15%, after removing number of letters and words.

**Removal of low-level speech features** In both reading and listening conditions, we make the following observations from Fig. 13 (b) and (d): (1) Removing DiPhones features from Text-based language models results in major drop (more than 25%) compared to other low-level speech features. (2) Conversely, removal of phonological features results in a larger performance drop (more than 80% of the original performance) for speech-based language models than text-based models (10% drop of the original performance).

Overall, the alignment of speech-based models with late language regions is almost entirely due to low-level stimulus features, and not brain-relevant semantics.

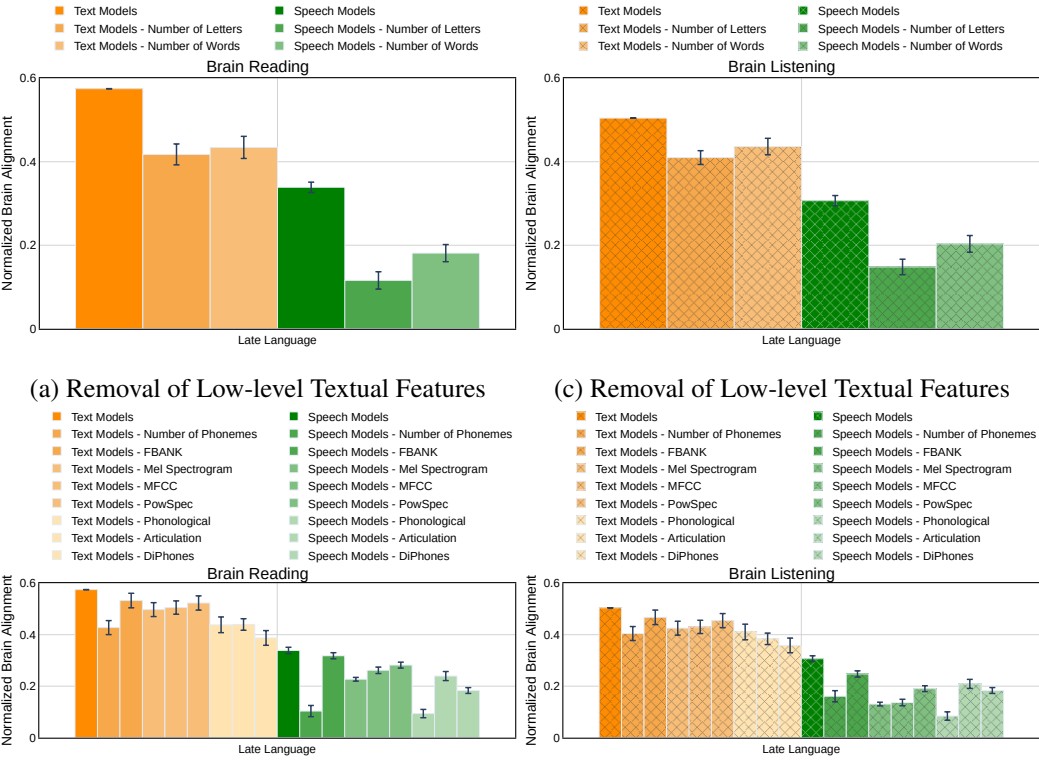

Figure 13: Reading condition in the Late Language: *(a)* low-level textual features, Average normalized brain alignment was computed over the average of participants for text and speech-based models, across layers for each low-level textual property. *(b)* low-level speech features, Average normalized brain alignment was computed over the average of participants for text and speech-based models, across layers for each low-level speech property. Listening condition in the Late Language: *(c)* low-level textual features: Average normalized brain alignment was computed across participants for text and speech-based models, across layers for each low-level text property. *(d)* low-level speech features, Average normalized brain alignment was computed over the average of participants for text and speech-based models, across layers for each low-level speech property.

# D LAYER-WISE PROBING ANALYSIS BETWEEN LANGUAGE MODELS AND LOW-LEVEL STIMULUS FEATURES

To investigate how much of the information in the low-level stimulus features can be captured by text- and speech-based language models, we learn a ridge regression model using model representations as input features to predict the low-level features (textual, speech and visual) as target. Fig. 16 shows that text-based language model (BERT) can accurately predict low-level textual features in

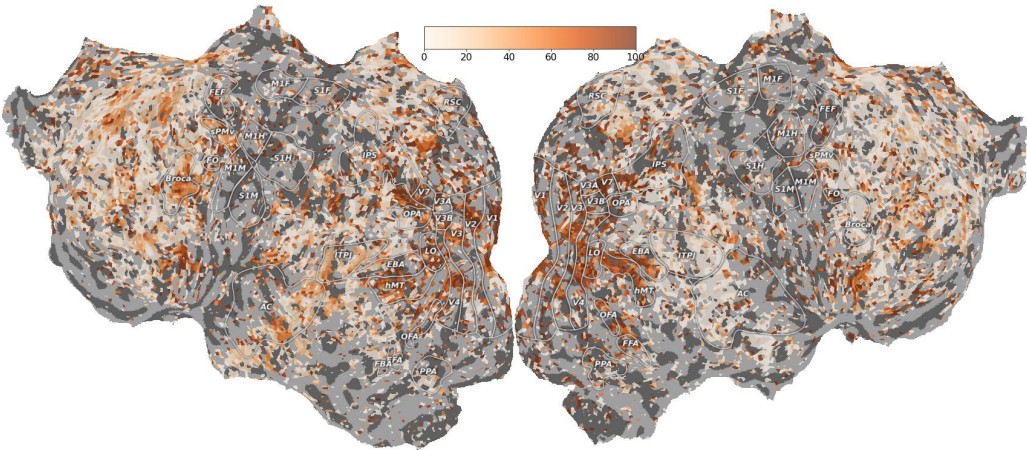

Figure 14: Reading: Percentage decrease in alignment for each voxel after removing number of letters feature from BERT representations. Percentage decrease scores for each voxel in one subject (subject-8) are projected onto the subject's flattened cortical surface. Voxels appear White if number of letters feature do not explain any shared information of BERT, and orange if all the information predicted BERT is similar to the information captured using number of letters feature. Voxels appear in light orange indicates that parts of late language explained (0-20% drop) by number of letters feature and BERT model has more information shared with late language regions beyond number of letters feature.

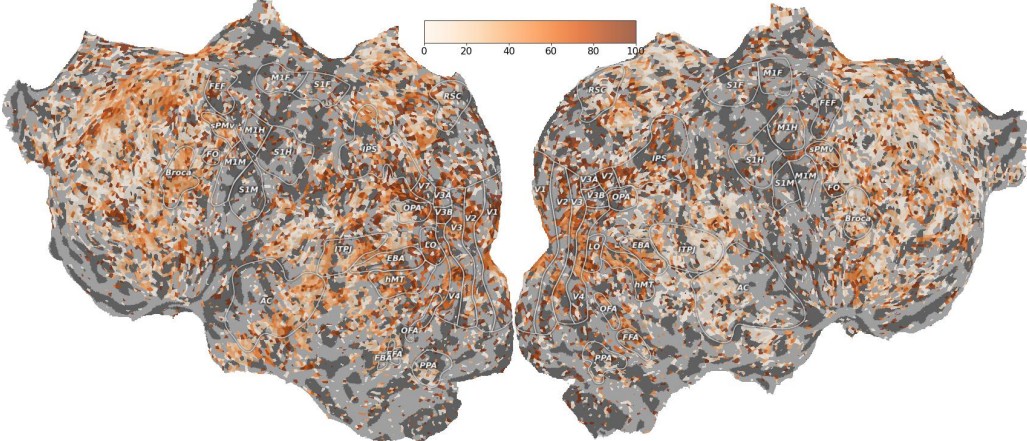

Figure 15: Brain Reading: Percentage decrease in brain alignment for each voxel by comparing the results after removing DiPhone features from BERT with the results before using BERT. Percentage decrease scores for each voxel in one subject (subject-8) are projected onto the subject's flattened cortical surface. Voxels appear White if DiPhone features do not explain any shared information of BERT, and orange if all the information predicted BERT is similar to the information captured using DiPhone features. Voxels appear in light orange indicates that parts of late language explained (20-40% drop) by DiPhone features and BERT model has more information shared with late language regions beyond DiPhone features.

the early layers and have decreasing trend towards later layers. For the low-level speech features, text-based models have zero to negative $R^2$-score values showing that text-based models do not have any speech-level information. We report GPT-2 probing results in the Fig. 16.

Complementary to text-based language models, speech-based models (Wav2Vec2.0) can accurately predict low-level speech features in the higher layers and have lower $R^2$-score values in the early layers.

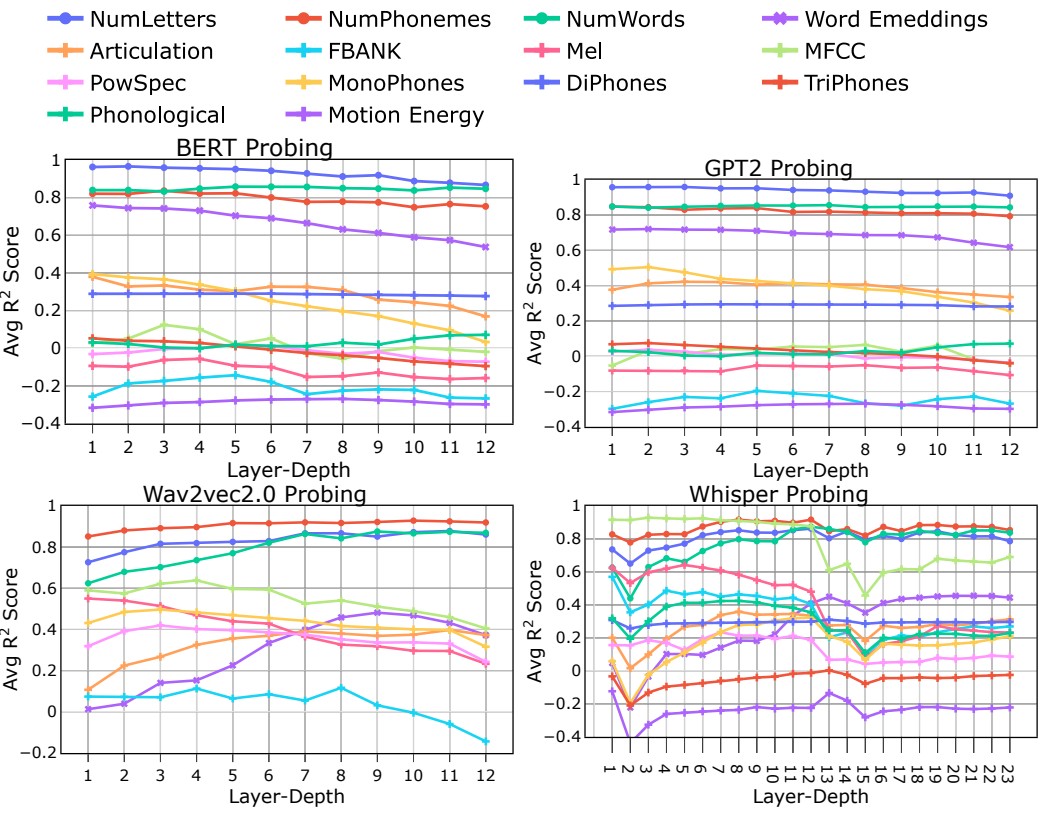

Figure 16: Probing the information (basic linguistic and speech features) represented across layers in neural language (BERT and GPT2) and speech-based models (Wav2Vec2.0 and Whisper).

## E  A1 REGION: SPEECH-LANGUAGE MODEL ALIGNMENT DURING LISTENING

We now show the results per speech model in the Fig. 17. During listening, specifically for the A1 region, we observe that both speech models, Wav2Vec2.0 and Whisper, exhibit high normalized brain alignment. Additionally, the elimination of low-level textual and speech features results in a significant performance decline in both models. However, it is important to note that these language models have additional information other than low-level features that need to be explored to further explain the early auditory region.

Similar to the A1 region, we observed that both Wav2Vec2.0 and Whisper exhibit similar normalized brain alignment in late language regions. Moreover, the removal of low-level textual and speech features results in a significant performance decline in both models.

## F  LAYER-WISE NORMALIZED BRAIN ALIGNMENT

We now plot the layer-wise normalized brain alignment for the Wav2Vec2.0 model in brain listening, both before and after removal of one important low-level speech property: phonological features, as shown in Fig. 18. Observation from Fig. 18 indicates a consistent drop in performance across layers, after removal of Phonological features, specifically in A1 and Late language regions. The key finding here is that our results that low level features impact the ability to predict both A1 and late language regions hold across individual layers.

## G  A1 REGION: LOW-LEVEL STIMULUS FEATURES AND BRAIN ALIGNMENT

We now plot the average normalized brain alignment for low-level stimulus features (textual, speech and visual) during both reading and listening in the early sensory areas (early visual and A1), as

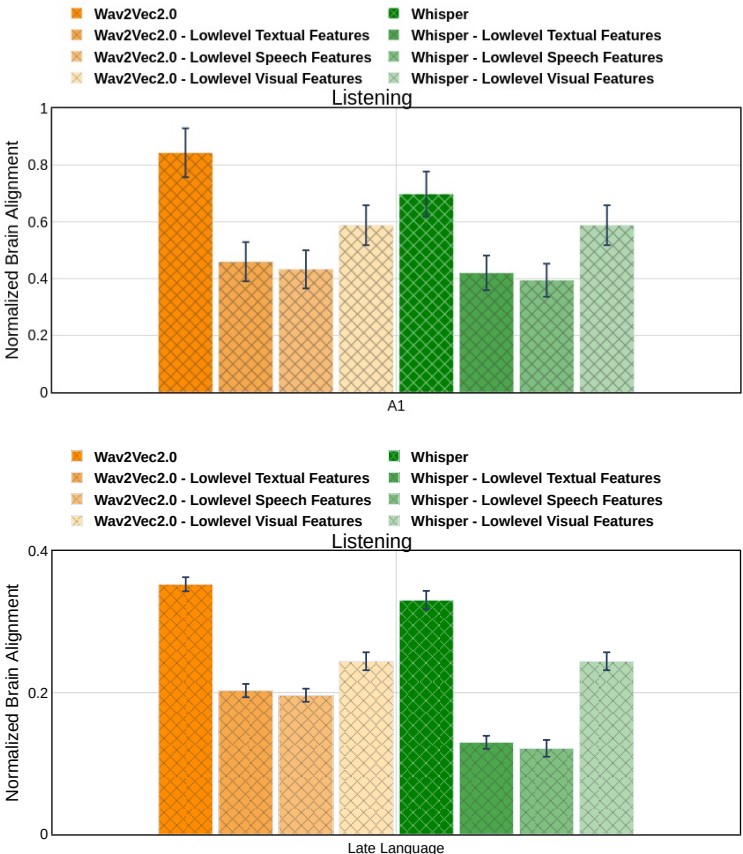

Figure 17: Brain Listening in A1 and Late language regions: Average normalized brain alignment was computed over the average of participants for speech-based models, across layers and across low-level features.

shown in Fig. 19. Additionally, we report the individual low-level stimulus features, such as the number of letters, PowSpec, Phonological features and motion energy features, specifically in early sensory processing regions. It appears that both text-based and speech-based language models meet the baselines in early sensory processing regions, particularly early visual areas in reading and A1 areas during listening. Among low-level stimulus features, motion energy features have better normalized brain alignment during reading in the early visual area and Phonological features have better brain alignment during listening in the A1 region.

## H    DISCUSSION AND CONCLUSION

We implement a direct approach to evaluate what types of information language models truly predict in brain responses. This is achieved by eliminating the information related to specific low-level stimulus features (textual, speech, and visual) and observing how this intervention affects the alignment with fMRI brain recordings acquired while participants read versus listened to the same naturalistic stories. We show that both text- and speech-based language models predict brain responses well both during reading and listening in corresponding early sensory regions in large parts due to low-level stimulus features that are correlated between text and speech (e.g., number of letters and phonemes). We also found that text-based models predict fMRI recordings significantly better than speech-based models in the late language regions, irrespective of stimulus modality.

Our findings have direct implications for both machine learning and cognitive neuroscience. First, we show that even during speech-evoked brain activity (i.e., listening), the alignment of speech-based models trails behind that of text-based models in the late language regions. More importantly, our results demonstrate that the alignment of speech-based models with these regions is almost

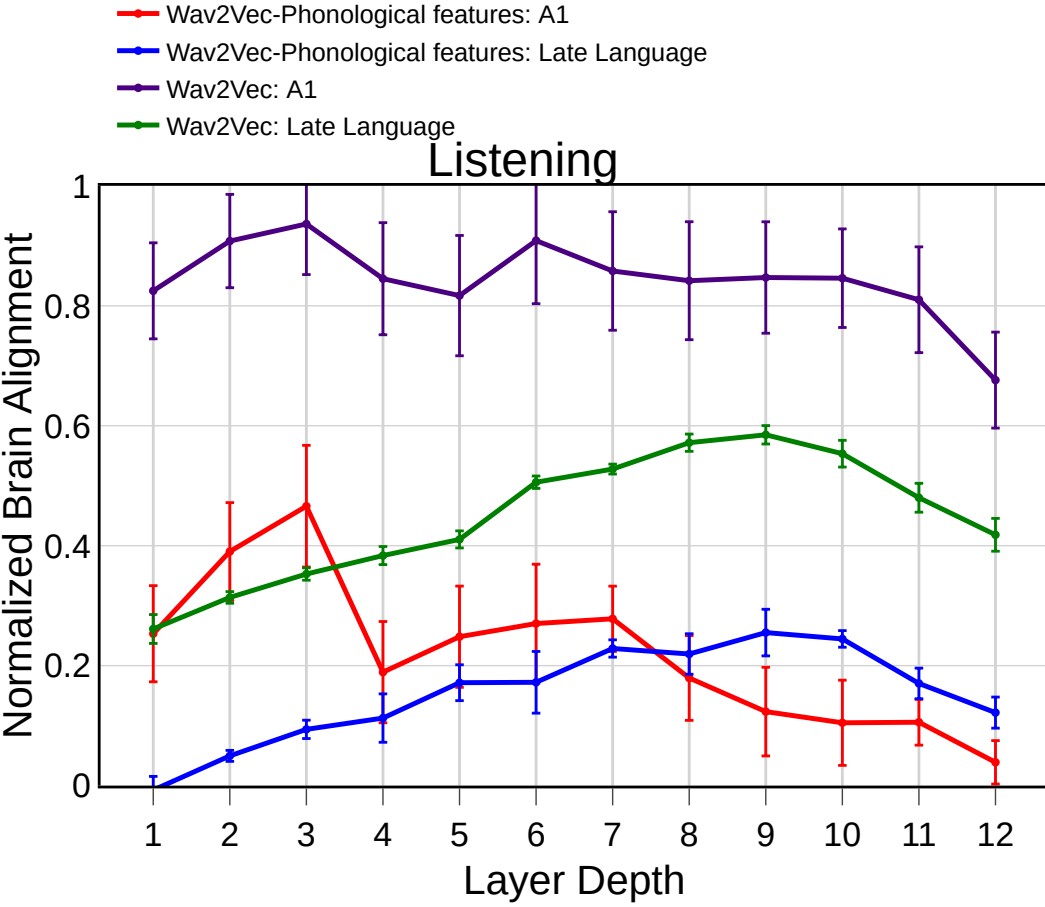

Figure 18: Layer-wise average normalized brain alignment was computed over the average of participants for speech-based model: Wav2Vec2.0 for an important low-level speech property: phonological features.

entirely explained by low-level stimulus features. Since these regions are purported to represent semantic information, this finding implies that contemporary speech-based models lack brain-relevant semantics. This suggests that new machine learning approaches are needed to improve speech-based models. Furthermore, these results imply that observed similarities between speech-based models and brain recordings in the past (Vaidya et al., 2022; Millet et al., 2022) are largely due to low-level information and not semantics, which is important to take into account when interpreting the similarity between language representations in speech-based models and the brain. Second, we observe that phonological features explain the most variance during listening for speech-based models, whereas "number of letters" explains the most variance during reading for text-based models. This result offers us a glimpse into the mechanisms underlying language processing in the brain. Third, we demonstrate a direct residual approach to identify the contribution of specific features to brain alignment. To our knowledge, there is no better alternative to selectively remove information from language models to probe their impact on brain alignment. Using this approach, it is possible to investigate how the human brain processes language during both reading and listening at a finer scale than before.

One limitation of our approach is that the removal method we use only removes linear contributions to language model representations. While this is sufficient to remove the effect on the brain alignment, which is also modeled as a linear function, it is possible that it does not remove all information related to the specific features from the model. Another possible limitation for the interpretation of the differences between the brain alignment of text- vs speech-based models is that the models we are using have several differences beside the stimulus modality, such as the amount of their training data and objective functions. To alleviate this concern, we have tested multiple models of each type,

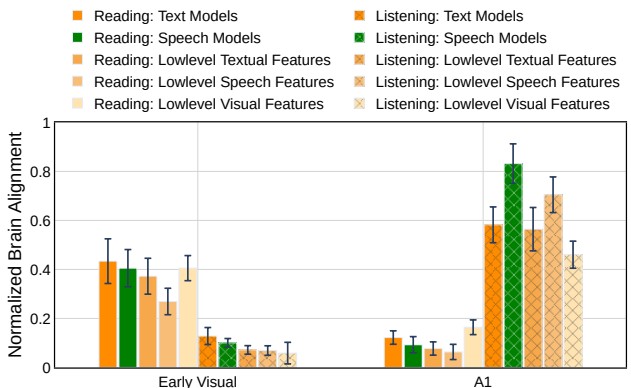

(a) Average of normalized brain alignment across low-level stimulus features

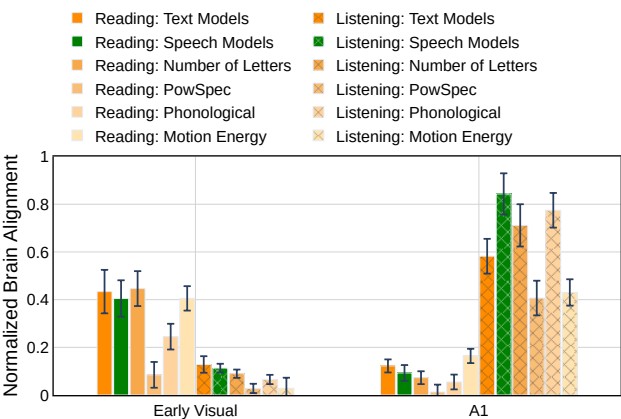

(b) Average of normalized brain alignment over individual low-level stimulus features

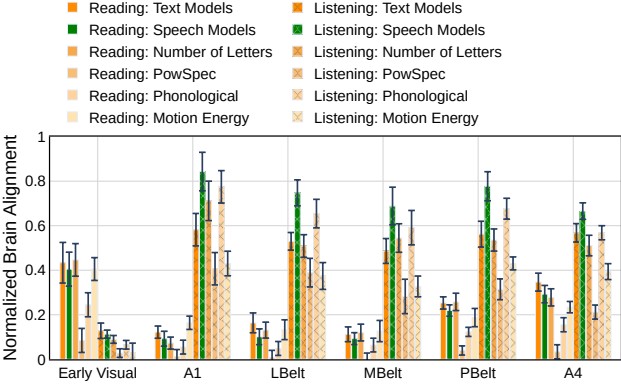

(c) Average of normalized brain alignment over individual low-level stimulus features. Here, Speech Models refer to only Wav2Vec2.0 model.

Figure 19: Early sensory processing regions: Average normalized brain alignment was computed over the average of participants across text-based and speech-based language models, along with basic low-level stimulus features, such as the number of letters, PowSpec, Phonological features and motion energy features.

with different objective functions and trained on different amounts of data, and showed that the results we observe generalize within the text- and speech-based model types despite these differences. Still, it is possible that some of the differences in brain alignment we observe are due to confounding differences between model types, and there is value in investigating these questions in the future with models that are controlled for architecture, objective, and training data amounts. Lastly, our work uses brain recordings that are obtained from English-speaking participants and experimental stimuli that are in English, and therefore we use models that are mostly trained on English text and speech. It is possible that the findings would differ in other languages, and this is important to study in the future.

The alignment of text-based models with the late language regions is not explained by low-level stimulus features alone. However, these regions also process high-level semantic information (e.g., discourse-level or emotion-related) (Binder & Desai, 2011; Wehbe et al., 2014; Bookheimer, 2002). Future work can investigate the contribution of such features to this alignment. In addition, while impressive, the current level of alignment does not reach the estimated noise ceiling. Inducing brain-relevant bias can be one way to enhance the alignment of these models with the human brain (Schwartz et al., 2019). Overall, further research is necessary to improve both text- and speech-based language models.

## I    ESTIMATED NOISE CEILING ACROSS ROIS

We present the average noise ceiling estimate across subjects for both reading and listening conditions in the Fig. 20. In this Fig. 20, we report the average noise ceiling estimate of all the voxels in every ROI.

One potential concern is that because the noise ceiling in the sensory regions of the non-presentation modality is low (A1 during reading, and V1 during listening), one may expect that using this low noise ceiling to normalize the prediction performance of models may result in an overly inflated number. However, from Fig. 20, we see that the normalized brain alignment in the sensory regions of the non-presentation modality is in fact quite low (Fig 1: checkered bars in early visual, and solid bars in early auditory).

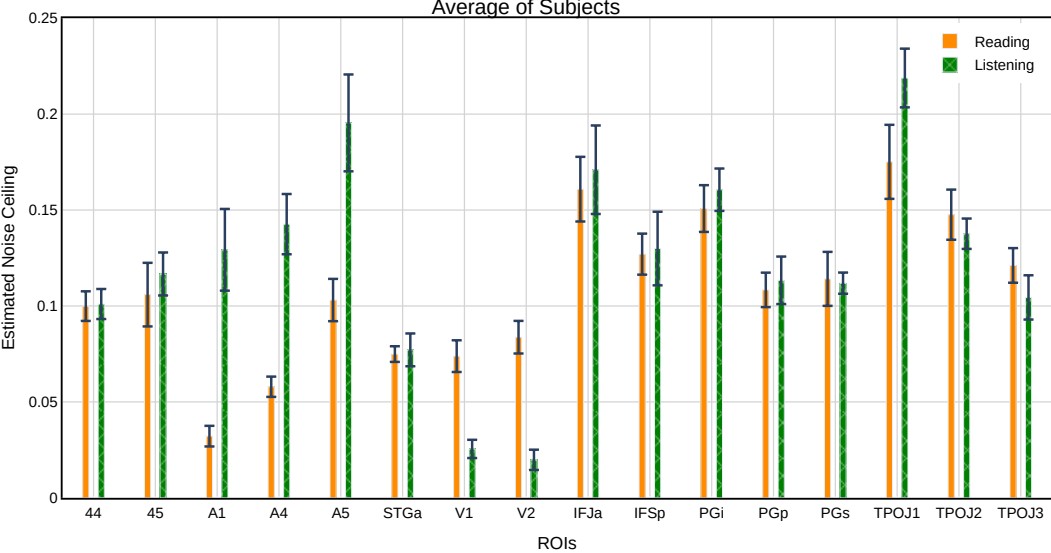

Figure 20: Layer-wise average normalized brain alignment was computed over the average of participants for speech-based model: Wav2vec2.0 for an imporant low-level speech property: phonological features.

## J    BRAIN LISTENING IN A1 REGION: TEXT-BASED VS. SPEECH-BASED LANGUAGE MODELS

We now show the results just for A1 region in Fig. 21. During listening, specifically for the A1 region, we observe that speech-based language models have higher normalized brain alignment than text-based models. Additionally, removal of low-level textual features results in a similar performance drop for both types of models. Also, removal of low-level speech features results in a larger performance drop for speech-based compared to text-based language models.

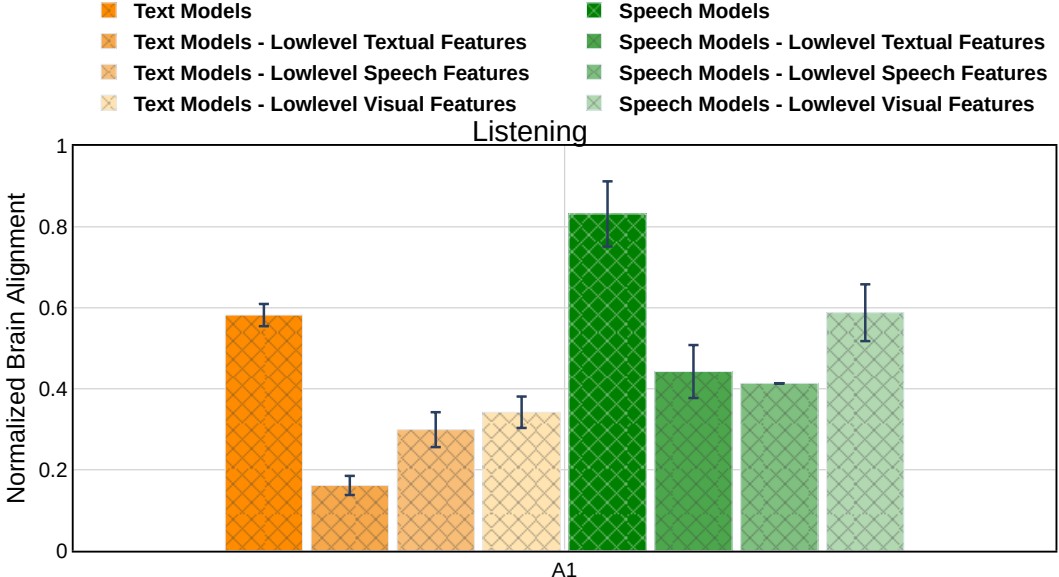

Figure 21: Brain Listening: (a) Removal of Low-level textual features, Average normalized brain predictivity was computed over the average of participants for text and speech-based models, across layers for each low-level textual property. (b) Removal of low-level speech features: Average normalized brain predictivity was computed across participants for text and speech-based models, across layers for each low-level speech property.

| Early visual | The early visual area is the earliest cortical region for visual processing. It processes basic visual features, such as edges, orientations, and spatial frequencies (). Lesions in V1 can lead to blindness in the corresponding visual field. V2 processes more complex patterns than V1. |
|---|---|
| VWFA | The visual word form area specializes in recognizing written words and letters, facilitating the transition from visual representations of words to their associated meanings and sounds (Dehaene & Cohen, 2011; McCandliss et al., 2003). This region is crucial for skilled reading. |
| Early auditory | The early auditory area is the earliest cortical region for speech processing. This area is specialized for processing elementary speech sounds, as well as other temporally complex acoustical signals, such as music. |
| Late Language | Late language regions contribute to various linguistic processes. Areas 44 and 45 (Broca's area) are vital for speech production and grammar comprehension (Friederici, 2011). The IFJ, PG, and TPOJ clusters are involved in semantic processing, syntactic interpretation, and discourse comprehension (Deniz et al., 2019; Toneva et al., 2022). STGa and STS play roles in phonological processing and auditory-linguistic integration (Vaidya et al., 2022; Millet et al., 2022; Gong et al., 2023). TA2 is implicated in auditory processing, especially in the context of language. |

Table 1: Detailed functional description of various brain regions.

