# OpenReview forum: "Speech language models lack important brain-relevant semantics"
_ICLR.cc/2024/Conference — Submitted to ICLR 2024_

### Official Review · Reviewer_jzFj · 2023-10-30

**Soundness:** 3 good
**Presentation:** 3 good
**Contribution:** 3 good
**Rating:** 5
**Confidence:** 4

**Summary:**

The authors present a detailed analysis predicting fMRI activity from the activations of text-based and speech-based neural network models. The fMRI dataset studied contains data obtained during both reading and listening to the same story, allowing the authors to directly compare how each type of model can predict a given modality. The authors build a regression model between low-level features and the model activations and use the residuals of this model to also predict the neural responses, which will evaluate how much of the prediction is due to correlations of the stimuli with low-level features encoded by the model. The authors demonstrate that although early visual areas can be predicted by the responses of such language networks, this is primarily driven by low-level cues. Similarly, although early auditory areas are somewhat predicted by the text-based model features, this is due to the low-level cues.

**Strengths:**

The dataset that the authors use provides a compelling way to get at the question of whether the features of language models trained on one modality (speech or text input) are able to predict the other modality). The authors claim that prior work presented puzzling evidence of language models predicting early sensory regions of a modality that they were not trained on, and this work provides evidence that this is primarily due to features that are correlated between the two modalities. The methodology is also very clearly described and well-documented.

**Weaknesses:**

W1: The paper lacks a general discussion about correlated features in natural stimuli, which, in my view, is directly what the authors are trying to test. Citing some past work along these lines would be helpful (for instance, Norman-Haignere et al. 2018, Groen et al. 2018).

W2: In various parts of the paper, there seems to be a bit of a conflation with correlation and causation. For instance, the authors state: “This raises questions about what types of information text-based language models capture that is relevant to early auditory processing.”, but I think the authors mean something like “This raises questions about what types of information text-based language models capture that is **correlated with features** relevant to early auditory processing”. Another instance: “This is possibly because the language models do not process visual features such as edges.”->“This is possibly because the **features in language models are not correlated with visual stimulus features** such as edges.” There are other places like this in the paper that I encourage the authors to fix.

W3: The title of the paper seems a bit limited in scope and focuses on a result in the paper that is a bit of a straw man. The speech-based models that were tested (wave2vec2.0 and whisper) are not trained to capture semantic information, so it doesn’t seem surprising that they fail to capture semantic-driven responses.

**Questions:**

Q1: In the first paragraph of the introduction, which of the cited papers claims that text-trained language models predict early sensory cortex activities? This is currently unclear (many of the cited works listed under “speech” only study models with a waveform or spectrogram input), and seems very important to distinguish as is the main motivation for the study.


Q2: Are there some baseline models that the authors could include to better contextualize the results? For instance, how well does a random-feature baseline do (useful to see a “lower bound”)? How well does a classic primary-area model such as the motion energy model, or a spectrotemporal filter bank do at capturing the primary area responses? For the classic model comparison, if neither language model is predicting the voxels better than these baseline models, then should we be considering the variance they are explaining as significant at all?


Q3: Is ridge regression necessary when fitting the low-level features to the neural model representations? It seems like there is nearly infinite available data for this fit (I believe the low-level features are automatically extracted) and there is no noise, so I’m just wondering what the intuition is for using a ridge parameter here.


Q4: Are the visual words and the onset of auditory words aligned in time in the dataset? That seems particularly important for this comparison, as one of the overall features encoded by both models may be “when a new thing starts”.

Minor points:

* The ROIs are described as “language relevant” but early visual and early auditory areas are included. It seems non-standard to refer to these as “language relevant”.

* There is a sentence about the estimated noise ceiling being based on an assumption of a perfect model. Is the “perfect model” referred to here the regression model used for the participant-particiant regression? Or is this “perfect model” one of the candidate encoding models from DNN predictions? Clarifying what this sentence means would help the reader.

---

> ### Author Response · Authors · 2023-11-22
>
> *We thank the reviewer for their insightful and valuable comments and suggestions which are crucial for further strengthening our manuscript.*
>
> **Q1: In the first paragraph of the introduction, which of the cited papers claims that text-trained language models predict early sensory cortex activities? This is currently unclear (many of the cited works listed under “speech” only study models with a waveform or spectrogram input), and seems very important to distinguish as is the main motivation for the study.**
>
> Thank you for the question.
> * We want to clarify that all cited papers related to text-based language models predict early sensory activity to an impressive degree. It is important to note that these studies used secondary auditory areas that are also part of language regions.
> * In case of speech-based language models, [Vaidya et al. 2022 ICML, Millet et al. 2022 NeurIPS] have shown that all SSL based models predict the early auditory cortex (A1 region) to an impressive degree and trails behind simple word embeddings across the rest of the cortex.
>
> *[Vaidya et al. 2022] Self-supervised models of audio effectively explain human cortical responses to speech, ICML-2022*
>
> *[Millet et al. 2022] Toward a realistic model of speech processing in the brain with self-supervised learning. NeurIPS-2022*
>
> **Q2: Are there some baseline models that the authors could include to better contextualize the results?
> How well does a classic primary-area model such as the motion energy model, or a spectrotemporal filter bank do at capturing the primary area responses? For the classic model comparison, if neither language model is predicting the voxels better than these baseline models, then should we be considering the variance they are explaining as significant at all?**
>
> * Based on the reviewer’s suggestion, we now plot the average normalized brain alignment for classical models, such as low-level stimulus features (textual, speech, and visual) during both reading and listening in the early sensory areas (early visual and A1), as shown in **Fig. 19 (see section G in Appendix)**.
> * Additionally, we report results for the individual low-level stimulus features as baseline models, including the number of letters, PowSpec, phonological features, and motion energy features, particularly in early sensory processing regions. It appears that both text-based and speech-based language models meet the baselines and show improvement in early sensory processing regions, particularly early visual areas in reading and A1 areas during listening.
> * Among low-level stimulus features, motion energy features have better normalized brain alignment during reading in the early visual area and phonological features have better brain alignment during listening in the A1 region.
>
> * Overall, in the context of classic model comparison, language model representations predict better than baseline models and the variance explained by these models is significant.
>
> **Q3. Is ridge regression necessary when fitting the low-level features to the neural model representations?**
>
> * We fit the removal method using the stimulus text, which is limited by the fMRI dataset. This is why we use ridge regression for the fitting procedure to minimize possible overfitting.
> * The reviewer’s suggestion to fit a more general removal approach using additional text data is very interesting to explore in future work
>
> **Q4. Are the visual words and the onset of auditory words aligned in time in the dataset? That seems particularly important for this comparison, as one of the overall features encoded by both models may be “when a new thing starts”.**
>
> * The same stories from listening sessions were used for reading sessions. Praat’s word representation for each story (W, t) was used for generating the reading stimuli. The words of each story were presented one-by-one at the center of the screen using a rapid serial visual presentation (RSVP) procedure.
> * During reading, each word was presented for a duration precisely equal to the duration of that word in the spoken story. RSVP reading is different from natural reading because during RSVP the reader has no control over which word to read at each point in time.
> * Therefore, to make listening and reading more comparable, the authors matched the timing of the words presented during RSVP to the rate at which the words occurred during listening.

---

> > ### Author Response · Authors · 2023-11-22
> >
> > **W1. The paper lacks a general discussion about correlated features in natural stimuli, which, in my view, is directly what the authors are trying to test. Citing some past work along these lines would be helpful (for instance, Norman-Haignere et al. 2018, Groen et al. 2018).**
> >
> > Thank you for suggesting relevant works on correlated features in natural stimuli. These citations will be incorporated in the updated draft. Further, we summarize the implications of our findings in the updated manuscript **(see Appendix section H)**.
> >
> > **W2.  In various parts of the paper, there seems to be a bit of a conflation with correlation and causation.**
> >
> > As per the reviewer's suggestion, we have now addressed the two sentences and other places in the updated draft.
> >
> > **W3. The title of the paper seems a bit limited in scope and focuses on a result in the paper that is a bit of a straw man. The speech-based models that were tested (wave2vec2.0 and whisper) are not trained to capture semantic information, so it doesn’t seem surprising that they fail to capture semantic-driven responses.**
> >
> > * All models that we test, both speech-based and text-based, are trained to predict a missing part of the input.
> > * Moreover, one of the speech-based models that we test (Whisper) has an encoder-decoder architecture and is trained as a speech-to-text model, so the objective function is quite close to that of a text-based language model.
> > * When text-based language models first came out, it was indeed surprising that they were able to perform so many semantic tasks despite being trained with such a simple objective function.
> > * However, we continue to see their performance excel and we also show here that their ability to predict many language regions is not due to low-level features.
> > * Even though speech-based models are trained in a similar way, we see that their ability to predict late language regions is almost entirely due to low-level features, suggesting that they fail to capture brain-relevant semantics in language regions.
> >
> > **Minor Points**
> >
> > **The ROIs are described as “language relevant” but early visual and early auditory areas are included.**
> >
> > Thank you for this suggestion. Now, we refer to the regions as: three early sensory and four language-relevant ROIs in the human brain.
> >
> > We updated these minor points in the revised draft.

---

> > > ### Comment · Reviewer_jzFj · 2023-11-23
> > > **Response to authors**
> > >
> > > Thank you for the clarifications and the text changes. After reading the responses and other reviewers' concerns I am deciding to keep my score as-is. This is partially due to the noise-ceiling discussion, which was also my reasoning for asking about the baseline models. Its overall difficult to interpret what the main results are.
> > >
> > > From reading the responses, one thing I'm noticing is that there seems to be a disagreement about what a 'language model' is. This is why I pointed out that the title of the paper seems like a straw man, and (at least from what I could tell) not the main focus of the paper (which seems to be asking questions about text-based models as well)? Calling wav2vec a language model definitely seems incorrect to me. As the authors note, whisper is a bit different because it does have an audio-conditional language model as a decoder, but it is still typically considered an ASR model so its just a bit non-standard, and I think it is an open question if that decoder captures semantic information.
> > >
> > > Overall, I think that the analyses presented in the paper seem interesting and promising, but need a bit more polishing to fully guide readers through their significance and also validate that the analysis methods are correct for the setting.

---

> > > > ### Author Response · Authors · 2023-11-23
> > > >
> > > > We thank you for engaging with us in a discussion. We want to take the opportunity to clarify the concerns you voice further:
> > > >
> > > >
> > > > ## Wav2vec as a language model:
> > > >
> > > > Speech-based models are often referred to as language models, because the general definition of a “language model” involves predicting the next token or sequence of tokens, given context. Even in the paper which introduces the wav2vec model we use [Baevski et al. 2020 NeurIPS], the authors state that the model is "pre-trained by masking specific time steps in the latent feature space, similar to the masked language modeling approach in BERT". Although the wav2vec model does not generate text in the way traditional text-based language models do, it plays a crucial role in processing and understanding spoken language, making it a type of speech language model. Additionally, in another work that introduces another popular speech model HuBERT, the authors explicitly state that the "HuBERT model is compelled to learn both acoustic and language models from continuous inputs" [Hsu et al. 2021 TASLP].
> > > >
> > > > Whether these speech models capture the same meaning of language as traditional text-based language models is not known. Cognitive neuroscientists are starting to study the overall alignment of such speech models with the human brain, across the whole cortex and not just in early auditory areas, so our work is extremely timely in showing that the observed alignment in non-sensory regions is largely due to low-level features.
> > > >
> > > > ## Interpretation of the main results:
> > > >
> > > > We are now updating the baseline results in Fig 19c now to reflect all subregions of the auditory cortex, in addition to A1, and are presenting only results from one of the speech models (wav2vec), to keep the setting consistent with Fig 6 in the main text. Note that wav2vec outperforms all low-level feature baselines significantly at predicting auditory cortex regions (paired t-test, pvalue < 0.05).
> > > >
> > > > Overall, we summarize our key results here, which were pointed out as valuable and as a good contribution by Reviewer cgRE:
> > > >
> > > > 1. The impressive predictive performance of speech models in the early auditory cortex is NOT entirely accounted for by low-level features. This is shown in Fig 6. As Reviewer cgRE kindly pointed out, we investigate a comprehensive set of low-level features that are thought to relate to processing in the early auditory cortex. We show that, surprisingly, there is additional explainable variance in many voxels in early auditory cortex is left to explain after removing the low-level features we consider. This shows that speech-based models capture additional features that are important for early auditory cortex and understanding these further can help us better understand this part of the brain. The baseline results that we produced in response to the reviewer also help with strengthening this points.
> > > > 2. Another surprising result is that, in contrast to the alignment of speech-based models in the early auditory cortex, the alignment in late language regions is almost entirely due to low-level features. The surprise is coming from the fact that the predictive performance of the late language regions is still good and almost on par with text-based models, and also from the fact that several recent works have proposed speech-based models for good models of language in the brain [Li et al. 2023 Nature Neuroscience, Chen et al. 2023 Arxiv]. Our work clearly demonstrates the importance of careful consideration of the reasons behind the alignment that is observed between models and human brain recordings.

---

### Official Review · Reviewer_cgRE · 2023-10-31

**Soundness:** 3 good
**Presentation:** 3 good
**Contribution:** 2 fair
**Rating:** 5
**Confidence:** 4

**Summary:**

This paper describes a re-analysis of two datasets of fMRI responses of a story from the Moth Radio Hour. In one cases, subjects listened to the story and in the other case the subjects read the story. The authors attempt to predict responses to these stories using text-based, large language models (BERT, GPT2, FLAN-T5) and audio-based, speech models (wav2vec2.0 and Whisper) using standard, regression-based voxelwise encoding models. They compare the prediction accuracy of these models with variants where they have regressed out the contribution from text-based features, audio- and speech-based features, and low-level visual features. They find that text- and speech-based models show similar overall prediction accuracy in early visual and early auditory regions, while text-based models show superior performance in putatively higher-level visual and language regions. They find that the performance of text-based models is relatively robust in higher-level language regions, maintaining relatively good performance when controlling for text, audio/speech, and visual features, consistent with a response to higher-level linguistic properties. In early sensory areas, there is a greater impact of controlling for these features suggesting that these lower-level features predict more of the response variance, as expected. The trends for the speech model are mostly similar, with the biggest difference being that the controlling for audio and speech features hurts performance more in high-level, language regions, suggesting that these models are not predicting high-level, linguistic properties.

**Strengths:**

I think directly comparing text and speech models is a valuable contribution to the literature.

The dataset investigated, with matched responses to reading and listening, is interesting and relevant to the questions being addressed.

I think there is value in highlighting the problem of feature correlations, which this paper does well.

They test a large set of control features. Their controls are more comprehensive than most papers I have seen.

**Weaknesses:**

The conclusion that lower-level features explain a large portion of the variance in lower-level sensory areas is not surprising. The fact that high-level language regions are more robust to lower-level features in the context of text-based language models is also not surprising, since the response is higher-level and the model does not have any visual or auditory information baked into it. The fact that speech models are impacted by including audio and speech features is not as surprising as the authors suggest, since the models are taking audio as input and the representations are only being derived from 2-second stimuli and thus lose much of the higher-level language information that is present in a text-based model. This paper is essentially confirmatory and should be framed as such, in my opinion.

I don’t see the benefit of the “direct” approach compared with measuring the unique variance of different feature sets using some kind of a variance partitioning framework such as that promoted by the Gallant and Huth labs. Conceptually, the main thing one wants to know is what fraction of the neural response variance can be uniquely explained by a particular model and how much is shared with other models. The direct approach seems like an indirect way to address that question. I also find the term “direct” unclear? What is direct about it? What would the indirect approach be?

There is no attempt to understand whether text- and speech-based models account for shared or unique variance from each other, which seems important and natural in the context of this paper.

There needs to be more detail on the speech models. Minimally, there needs to be a summary of the tasks (e.g., masked token prediction) they were trained on and the maximum possible temporal extent that the models are able to consider. If possible, the authors should extend the window they consider to go beyond 2 seconds to allow the models to potentially incorporate longer timescale linguistic information.

Averaging performance across models is suboptimal because some of the models might be performing quite well, which would be valuable to know. For example, Whisper has been trained on a much broader range of tasks than wav2vec2.0 and it would be useful to know whether it performs better as a consequence. A better choice would be to select the best performing model for the main figure and to put the performance of all models in the appendix. Model selection could be done on training or validation data to prevent overfitting.

It is unclear how activations from different layers were handled. Were they all combined together? Typically, one selects the best-performing layer in a model using the training or validation set.

The authors need more detail about the stimuli. They should specify the total duration of the story(ies) in the listening and reading conditions, how many words there were, how the words were presented, and the rate they were presented at. For example, for listening, was this a natural story with a variable word rate? Or were the words presented artificially using a fixed ISI? If they used a variable word rate, how does this impact how the features were calculated? Downsampling does not seem straightforward in this case. For the reading condition, how was the text presented? Was there a word presented every few hundred milliseconds or was a whole sentence presented at once? Similarly how does this impact the feature design?

The language ROIs includes many regions of the STG that I would consider high-level auditory regions (e.g., respond similarly to native and foreign speech). I would recommend repeating the analyses with the language parcels released by the Fedorenko lab, or at least limiting yourself to the STS. For the early auditory analysis, I think it would be worth repeating these analyses with just the A1 parcel to be more conservative.

The visual word form area is quite small and challenging to localize:
https://www.pnas.org/doi/abs/10.1073/pnas.0703300104

I suspect the results here reflect what one would see from a generic high-level fusiform visual region. The authors could test this by selecting another nearby region and seeing if the results differ. If the results are similar, I think it is misleading to describe the results as specific to the visual word form area, despite the label provided by the atlas.

I could not follow how the noise ceiling is calculated. What is done with the results from all of the different subsamples? Is there some attempt to extend the results to infinite samples? I am skeptical about calculating a noise ceiling in V1 or A1 for the non-preferred modality. I would expect the noise ceiling to be very close to 0. How was this handled? When possible, it would be preferable to plot both the raw scores and the noise ceiling on the same figure so that you can see both. When you average across voxels for ROI analyses, do you average the noise-corrected values or do you separately average the raw and noise ceiling values and then divide these two numbers?

For Figure 3, it would be preferable to group by the listening/reading as was done in later figures. The performance between the modalities is not really comparable as these are totally different stimuli (and I am skeptical of the noise ceiling calculation).

The equations in the section title “Removal of low-level features from language model representations” make it seem like there is a single regularization term for all of the model features. It seems preferable to do what was done for the neural analyses and to fit a banded ridge model separately on every model feature. The different low-level features have very different dimensionality, so there should be some discussion of how this was handled when concatenating the features. If you z-score each feature than features that have higher-dimensionality will have much more influence. It was also not clear to me how cross-validation was handled here. Did you train and validate on a subset of stimuli and then remove the predicted response on test? How many folds were there? This information about cross-validation should also be specified in the voxel-wise encoding model section. For the banded ridge regression, were the lambdas specified separately varied for each feature set? How do we know that this range is sufficient? It is highly sensitive to the scale of the features. How fine was the grid search?

**Questions:**

In most cases, I found it easier to include my questions in the weaknesses section. See above.

What was the reason for constraining the text window to 20 words? How are the results impacted by this choice?

What is the reason for not removing the control features from the neural responses as well? How would doing so impact the results?

---

> ### Author Response · Authors · 2023-11-22
>
> *We thank the reviewer for their insightful and valuable comments and suggestions which are crucial for further strengthening our manuscript.*
>
> **Q. The influence of low-level features on early sensory processing regions and the resilience of high-level language regions in text-based models, are unsurprising. Similarly, the impact on speech models when incorporating audio features aligns with expectations, making the paper essentially confirmatory in nature.**
>
> It’s not clear which results about the speech models align with the reviewer’s expectations. We find the following quite surprising and noteworthy:
> * The impressive predictive performance of speech models in the early auditory cortex is NOT entirely accounted for by low-level features. As the reviewer kindly pointed out, we investigate a comprehensive set of low-level features that are thought to relate to processing in the early auditory cortex. We show that, surprisingly, 40-50% of the explainable variance in the early auditory cortex is left to explain after removing the low-level features we consider. This shows that speech-based models capture additional features that are important for early auditory cortex and understanding these further can help us better understand this part of the brain.
> * Another surprising result is that, in contrast to the alignment of speech-based models in the early auditory cortex, the alignment in late language regions is almost entirely due to low-level features. The surprise is coming from the fact that the predictive performance of the late language regions is still good and almost on par with text-based models, and also from the fact that several recent works have proposed speech-based models for good models of language in the brain [Yuanning Li et al. 2023 Nature Neuroscience, Chen et al. 2023 Arxiv]. Our work clearly demonstrates the importance of careful consideration of the reasons behind the alignment that is observed between models and human brain recordings.
>
> *[Yuanning Li et al. 2023] Dissecting neural computations in the human auditory pathway using deep neural networks for speech, Nature Neuroscience*
>
> *[Chen et al. 2023] Do self-supervised speech and language models extract similar representations as human brain?*
>
> **Q. I also find the term “direct” unclear? What is direct about it? What would the indirect approach be?**
>
> We thank the reviewer for this question and we will clarify this in the main paper.
>
> * An indirect approach is one which first relates model representations to the human brain, followed by independent examination of the related model to some task performance or behavioral output. For instance, [Schrimpf et al. 2021 PNAS] test the computations of a language model that may underlie human language understanding. This is accomplished by an independent examination of the relationship between the models’ ability to predict an upcoming word and their brain predictivity. Similarly, [Goldstein et al. 2022 Nature Neuroscience] provides empirical evidence that both the human brain and language model engage in continuous next-word prediction before word onset.
>
> * In contrast, the approach we use directly estimates the impact of a specific feature on the alignment between the model and the brain recordings by observing the difference in alignment before and after the specific feature is computationally removed from the model representations.  This is why we refer to this approach as direct.
>
> [Schrimpf et al. 2021]  The neural architecture of language: Integrative modeling converges on predictive processing. PNAS, 2021
>
> [Goldstein et al. 2022]  Shared computational principles for language processing in humans and deep language models. Nature neuroscience, 2022
>
> **Q. I don’t see the benefit of the “direct” approach compared with measuring the unique variance of different feature sets using some kind of a variance partitioning framework such as that promoted by the Gallant and Huth labs.**
>
> We hope that the response to the previous question clarified what we mean by a direct approach. The method we use to remove the linear contribution of a feature to a model’s representation is one way to implement such a direct approach. Another method was investigated by previous work [Oota et al. 2023 NeurIPS] and was shown to yield very similar results. Other methods can also be used. One can imagine devising a similar approach based on variance partitioning; however it is not immediately clear to us how to best take into account the estimate of the noise ceiling in the variance partitioning calculation.
>
> [Oota et al. 2023 NeurIPS] Joint processing of linguistic properties in brains and language models. NeurIPS-2023

---

> ### Author Response · Authors · 2023-11-22
>
> **Q. Whether text- and speech-based models account for shared or unique variance from each other?**
>
> * The direct comparison of text-based and speech-based language models in terms of their brain alignment is not straightforward to do in a rigorous way with models that are currently publicly available.
> * There are many differences between these types of models beyond the data modality, such as the amount of their training data and objective function. Instead of comparing these models that differ in many ways directly, we opt for comparing a model only to itself after removal of a particular feature.  This way we know exactly what a difference in brain alignment is due to: the removal of the specific feature.
> * We agree that directly comparing speech- vs text-based models is an exciting avenue for future work and we hope that it will be possible to do in a scientifically rigorous way soon. We summarize this discussion in the updated manuscript (see Appendix section H).
>
> **There needs to be more detail on the speech models. If possible, the authors should extend the window they consider to go beyond 2 seconds to allow the models to potentially incorporate longer timescale linguistic information.**
>
> Thank you for the question.
> * Previous work that considered longer windows (~64 seconds) for speech-based language models found that speech-based representations perform worse at predicting brain recordings outside of the primary auditory area than simple text-based word embeddings (without any context) [Vaidya et al. 2022 ICML, Millet et. al. 2022 NeurIPS].
> * This implies that even when longer timescales are considered, the speech-based representations do not substantially improve their ability to predict later language regions.  Still, there is room to examine the effect of context window using a direct approach similar to ours, and this would make a nice direction for future work.
>
> The details of two speech models as follow:
> |Model|Input|Training data size|Loss type|
> |-------|-------|-------|-------|
> |Wav2Vec2.0-base|Waveform|250K hours of raw speech|Masked contrastive loss|
> |Whisper-base|Log Mel spectrogram|680K hours of speech (raw speech+speech tasks)|Masked dynamic loss|
>
> [Vaidya et al. 2022] Self-supervised models of audio effectively explain human cortical responses to speech, ICML-2022
>
> [Millet et al. 2022] Toward a realistic model of speech processing in the brain with self-supervised learning. NeurIPS-2022
>
> **Q. Averaging performance across models is suboptimal because some of the models might be performing quite well: Wav2Vec2.0 vs. Whisper ?**
>
> We now show the results per model in the updated **Appendix F (see Fig. 17)**.
> * During listening, specifically for the A1 region, we observe that both speech models, Wav2Vec2.0 and Whisper, exhibit high normalized brain alignment. Additionally, the elimination of low-level textual and speech features results in a significant performance decline in both models. However, it is important to note that these language models have additional information other than low-level features that need to be explored to further explain the early auditory region.
> * Similar to the A1 region, we observed that both Wav2Vec2.0 and Whisper exhibit similar normalized brain alignment in late language regions. Moreover, the removal of low-level textual and speech features results in a significant performance decline in both models.

---

> > ### Author Response · Authors · 2023-11-22
> >
> > **Q. The authors need more detail about the stimuli.**
> >
> > We use the publicly available naturalistic story reading and listening fMRI dataset provided by [Deniz et al. 2019 Journal of Neuroscience]. The dataset consists of 11 stories, 6 participants, and all participants listened to and read all the stories. The speech stimuli consisted of 10- to 15 min stories taken from The Moth Radio Hour and used previously [Huth et al., 2016 Nature Neuroscience]. The 10 selected stories cover a wide range of topics and are highly engaging. The model validation dataset consisted of one 10 min story. All stimuli were played at 44.1 kHz using the pygame library in Python. The audio of each story was down-sampled to 11.5 kHz and the Penn Phonetics Lab Forced Aligner was used to automatically align the audio to the transcript. Finally the aligned transcripts were converted into separate word and phoneme representations using Praat’s TextGrid object. The word representation of each story is a list of pairs (W, t), where W is a word and t is the onset time in seconds.
> >
> > The same stories from listening sessions were used for reading sessions. Praat’s word representation for each story (W, t) was used for generating the reading stimuli. The words of each story were presented one-by-one at the center of the screen using a rapid serial visual presentation (RSVP) procedure. During reading, each word was presented for a duration precisely equal to the duration of that word in the spoken story. RSVP reading is different from natural reading because during RSVP the reader has no control over which word to read at each point in time. Therefore, to make listening and reading more comparable, the authors matched the timing of the words presented during RSVP to the rate at which the words occurred during listening.
> >
> > The total number of words in each story as follows:
> > |Story|Number of Words|
> > |-------|-------|
> > |Story1|2174|
> > |Story2|1469|
> > |Story3|1964|
> > |Story4|1893|
> > |Story5|2209|
> > |Story6|2786|
> > |Story7|3218|
> > |Story8|2675|
> > |Story9|1868|
> > |Story10|1641|
> > |Story11|1839 (test dataset)
> >
> > *[Huth et al. 2016], Natural speech reveals the semantic maps that tile human cerebral cortex. In Nature Neuroscience, 2016*
> >
> > *[Deniz et al. 2019], The representation of semantic information across human cerebral cortex during listening versus reading is invariant to stimulus modality. Journal of Neuroscience, 2019*
> >
> > **Q. Downsampling does not seem straightforward in this case.**
> >
> > Thank you for this question.
> > * We want to clarify that the same stories from listening sessions were used for reading sessions. Specifically, during reading, each word was presented for a duration precisely equal to the duration of that word in the spoken story. This implies that the downsampling procedure remains the same for both reading and listening conditions, and the downsampling is performed as follows:
> > * Since the rate of fMRI data acquisition (TR = 2.0045sec) was lower than the rate at which the stimulus was presented to the subjects, several words fall under the same TR in a single acquisition. Hence, we match the stimulus acquisition rate to fMRI data recording by downsampling the stimulus features using a 3-lobed Lanczos filter. After downsampling, we obtain the chunk-embedding corresponding to each TR.
> > * It is important to note that this downsampling approach is a well-established method to fit the fMRI TRs [Jain & Huth 2018 NeurIPS, Toneva & Wehbe 2019 NeurIPS, Deniz et. al. 2019 Journal of Neuroscience].
> > * We have already discussed this downsampling procedure in the Extracting text representations paragraph.
> >
> > *[Jain & Huth, 2018], Incorporating context into language encoding models for fmri. In NIPS-2018*
> >
> > *[Toneva & Wehbe 2019], Interpreting and improving natural-language processing (in machines) with natural language-processing (in the brain). In NeurIPS-2019*
> >
> > *[Deniz et al. 2019], The representation of semantic information across human cerebral cortex during listening versus reading is invariant to stimulus modality. Journal of Neuroscience. 2019*

---

> > > ### Author Response · Authors · 2023-11-22
> > >
> > > **Q. It is unclear how activations from different layers were handled. Were they all combined together?**
> > >
> > > Thank you for the question.
> > > * We want to clarify that each layer’s activations were used independently to build voxel-wise encoding models, and the results presented in the paper depict the average normalized brain alignment across all of the layers for both text-based and speech-based language models and their corresponding residuals. Based on the reviewers’ suggestion, we now plot the layer-wise normalized brain alignment for the Wav2Vec2.0 model during listening, both before and after removal of one important low-level speech property:  phonological features, as shown in **Fig. 18. (see in the updated Appendix G)**.
> > >
> > > * The results indicate a consistent drop in performance across layers, after removal of Phonological features, specifically in both A1 and the late language regions.
> > > * The key finding from this analysis is that our results that low level features impact the ability to predict both A1 and late language regions hold across individual layers.
> > >
> > > **Q. I would recommend repeating the analyses with the language parcels released by the Fedorenko lab**
> > >
> > > * We want to clarify that our selection of late language regions is in fact based on language parcels released by the Fedorenko lab, encompassing broader language regions. These regions include: (i) angular gyrus (AG: PFm, PGs, PGi, TPOJ2, TPOJ3), (ii) anterior temporal lobe (ATL: STSda, STSva, STGa, TE1a, TE2a, TGv, and TGd); (iii) posterior temporal lobe (PTL:  A4, A5, STSdp, STSvp, PSL, STV, TPOJ1); (iv) inferior frontal gyrus (IFG: 44, 45, IFJa, IFSp);  and (v) middle frontal gyrus (MFG: 55b). We will provide this clarification on these language parcels in the revised draft.
> > >
> > > **Q. For the early auditory analysis, I think it would be worth repeating these analyses with just the A1 parcel to be more conservative.**
> > >
> > > * We now show the results just for A1 region in the updated **Appendix section J (see Fig. 21)**.
> > > * During listening, specifically for the A1 region, we observe that speech-based language models have higher normalized brain alignment than text-based models.
> > > * Additionally, removal of low-level textual features results in a similar performance drop for both types of models.
> > > * Also, removal of low-level speech features results in a larger performance drop for speech-based compared to text-based language models.
> > >
> > > **Q. The visual word form area is quite small and challenging to localize. The authors could test this by selecting another nearby region and seeing if the results differ. If the results are similar, I think it is misleading to describe the results as specific to the visual word form area, despite the label provided by the atlas.**
> > >
> > > Thank you for this question.
> > > * We conducted the experiments as suggested by the reviewer and indeed found similar patterns in nearby regions, according to the atlas we are using.
> > > * Since we do not have functional localizers for the VWFA region, the obtained results are inconclusive because it may be that the nearby regions in the atlas overlap with the a particular participant's true VWFA.
> > > * Consequently, we will refer to these regions as fusiform gyrus areas in the paper instead of VWFA and say that VWFA is part of it but disclose that we are not able to localize it specifically.
> > >
> > > **Q. How the noise ceiling is calculated. What is done with the results from all of the different subsamples? Is there some attempt to extend the results to infinite samples?**
> > >
> > > * We follow previous work [Schrimpf et al. 2021 PNAS] to estimate the noise ceiling for the reading and listening fMRI datasets.
> > >  * We replicated the same noise ceiling estimate method for our two naturalistic story reading and listening datasets. Specifically, we create all possible combinations of s participants (s є [2,6]), separately for reading and listening. For each subsample s, we estimate the amount of brain response in one target participant that can be predicted using only data from other participants, using a voxel-wise encoding model.
> > > * In line with the noise ceiling estimate in previous work [Schrimpf et al. 2021 PNAS], we replicated the same code and extended the results to an infinite number of participants.
> > >
> > > *[Schrimpf et al. 2021],  The neural architecture of language: Integrative modeling converges on predictive processing. PNAS, 2021*

---

> > > > ### Author Response · Authors · 2023-11-22
> > > >
> > > > **Q. I am skeptical about calculating a noise ceiling in V1 or A1 for the non-preferred modality.
> > > > I would expect the noise ceiling to be very close to 0. How was this handled? When possible, it would be preferable to plot both the raw scores and the noise ceiling on the same figure so that you can see both.**
> > > >
> > > > Thank you for stressing this point.
> > > > * One potential concern is that because the noise ceiling in the sensory regions of the non-presentation modality is low (A1 during reading, and V1 during listening), one may expect that using this low noise ceiling to normalize the prediction performance of models may result in an overly inflated number.
> > > > * However, we see that the normalized predictivity in the sensory regions of the non-presentation modality is in fact quite low (Fig 1: checkered bars in early visual, and solid bars in early auditory). We hope that this clarification alleviates the reviewer's concerns.
> > > > * Based on reviewers suggestion, we present the average noise ceiling estimate across subjects for both reading and listening conditions in the **Appendix section I (see Fig. 20)**. In this Figure, we report the average noise ceiling estimate of all the voxels in every sub-ROI.
> > > >
> > > > **When you average across voxels for ROI analyses, do you average the noise-corrected values or do you separately average the raw and noise ceiling values and then divide these two numbers?**
> > > >
> > > > * The process for obtaining normalized brain alignment involved initially selecting voxels with a noise-ceiling estimate > 0.05, in line with previous works [Popham et al. 2021 Nature Neuroscience, La Tour et al. 2022, La Tour et al. 2023]. Subsequently, we divide each voxel prediction by the corresponding voxel’s noise ceiling estimate, and then average this normalized prediction across all selected voxels.
> > > >
> > > > *[Popham et al. 2021], Visual and linguistic semantic representations are aligned at the border of human visual cortex. Nature Neuroscience*
> > > >
> > > > *[La Tour et al. 2022 Neuroimage], Feature-space selection with banded ridge regression. NeuroImage.*
> > > >
> > > > *[La Tour et al. 2023], Voxelwise modeling tutorials: an encoding model approach to functional MRI analysis. Arxiv*
> > > >
> > > > **Q. What was the reason for constraining the text window to 20 words? How are the results impacted by this choice?**
> > > >
> > > > * Previous studies have examined the effect of amount of context on the brain predictivity [Jain and Huth, 2018 NIPS and Toneva and Wehbe, 2019 NeurIPS]. These works found a general increase in the ability to predict regions in the language network up to 20 words, and then there is a leveling off of the context effect. This is why we use 20 words. We will clarify this reasoning in the paper.
> > > >
> > > > *[Jain & Huth, 2018], Incorporating context into language encoding models for fmri. In NIPS-2018*
> > > >
> > > > *[Toneva & Wehbe 2019], Interpreting and improving natural-language processing (in machines) with natural language-processing (in the brain). In NeurIPS-2019*
> > > >
> > > > **Q. The performance between the modalities is not really comparable as these are totally different stimuli (and I am skeptical of the noise ceiling calculation).**
> > > >
> > > > * We want to clarify that the naturalistic reading and listening sessions involve exactly the same stories. Therefore, the naturalistic stimuli is the same for both modalities, enabling a direct comparison between reading and listening across different types of models.
> > > > * Moreover, the noise ceiling estimate is computed separately within each condition, and this is useful to measure the normalized brain alignment, particularly in case there is a difference in the signal-to-noise ratio between the two presentation modalities. Our observations reveal that in the late language regions, the noise ceilings are comparable, as shown in Fig2.
> > > >
> > > > **Q. “Removal of low-level features from language model representations” make it seem like there is a single regularization term for all of the model features.**
> > > >
> > > > Thank you for this question.
> > > > * It appears that there is some confusion about learning a linear function to map low-level features to language model representations, and we will clarify this methodology in the main paper. We reiterate that each feature corresponds to one regularization term during the learning of the linear function. Consequently, we fit a ridge model separately on every feature.
> > > >
> > > > **Q. The different low-level features have very different dimensionality, so there should be some discussion of how this was handled when concatenating the features.**
> > > >
> > > > * The low-level features indeed have different dimensionality, but each of the low-level features are handled independently and are not being concatenated. Note that the residual language model representations have the same dimensions across all low-level features, which are identical to the dimensionality of the language model representation.

---

> > > > > ### Author Response · Authors · 2023-11-22
> > > > >
> > > > > **Q. It was also not clear to me how cross-validation was handled here. Did you train and validate on a subset of stimuli and then remove the predicted response on test? How many folds were there? This information about cross-validation should also be specified in the voxel-wise encoding model section.**
> > > > >
> > > > > Thank you for this question.
> > > > > * To clarify, we first cross-validated on the training data set stimuli. Then for each fold in the banded ridge regression cross-validation, we trained the ridge regression model to remove low-level features to that particular fold. Overall, we performed a 10-fold cross-validation scheme as the training dataset stimuli consists of 10 different stories.
> > > > >
> > > > > We will clarify this cross-validation scheme in the revised draft.
> > > > >
> > > > > **Q. For the banded ridge regression, were the lambdas specified separately varied for each feature set? How do we know that this range is sufficient? It is highly sensitive to the scale of the features. How fine was the grid search?**
> > > > >
> > > > > * Our banded ridge regression method follows a similar training methodology used in previous brain encoding studies, including those by [ Deniz et al. 2019 Journal of Neuroscience, Vaidya et. al. 2022 ICML].
> > > > > * During the training of our neural encoding model, we varied the lambda parameters within the range of 10$^{-1}$ to 10$^{-3}$.
> > > > > * Finally, the range of lambdas selected for each feature set has been thoroughly tested, and our implementation aligns with encoding methods inspired by the work of the Gallant and Huth labs.
> > > > >
> > > > > *[Deniz et al. 2019], The representation of semantic information across human cerebral cortex during listening versus reading is invariant to stimulus modality. Journal of Neuroscience. 2019*
> > > > >
> > > > > *[Vaidya et al. 2022] Self-supervised models of audio effectively explain human cortical responses to speech, ICML-2022*
> > > > >
> > > > > **Q. What is the reason for not removing the control features from the neural responses as well? How would doing so impact the results?**
> > > > >
> > > > > * The features can be removed from the model representations (as we do), from the brain recordings, or from both (as suggested by the reviewer).
> > > > > * Conceptually, the results of these approaches should be the same because the feature is removed completely from either the input or/and the target and cannot further impact the observed alignment.
> > > > > * However, practically, brain recordings are noisier than model representations and so estimating the removal regression model will be more difficult especially with a limited sample size.
> > > > > * Therefore, we opt to remove the features from the model representations.
> > > > >
> > > > > We will clarify this reasoning in the main paper.

---

> > > > > > ### Comment · Reviewer_cgRE · 2023-11-22
> > > > > > **Comments on Contribution**
> > > > > >
> > > > > > Many thanks for the detailed and thoughtful reply to my comments.
> > > > > >
> > > > > > Some additional comments and questions.
> > > > > >
> > > > > > I felt I was pretty clear about what results aligned with my expectations:
> > > > > >
> > > > > > The conclusion that lower-level features explain a large portion of the variance in lower-level sensory areas is not surprising.
> > > > > >
> > > > > > The fact that high-level language regions are more robust to lower-level features in the context of text-based language models is also not surprising, since the response is higher-level and the model does not have any visual or auditory information baked into it.
> > > > > >
> > > > > > The fact that speech models are impacted by including audio and speech features is not as surprising as the authors suggest, since the models are taking audio as input and the representations are only being derived from 2-second stimuli and thus lose much of the higher-level language information that is present in a text-based model.
> > > > > >
> > > > > > What was unclear?
> > > > > >
> > > > > > You state in your rebuttal two key contributions:
> > > > > >
> > > > > > The impressive predictive performance of speech models in the early auditory cortex is NOT entirely accounted for by low-level features. As the reviewer kindly pointed out, we investigate a comprehensive set of low-level features that are thought to relate to processing in the early auditory cortex. We show that, surprisingly, 40-50% of the explainable variance in the early auditory cortex is left to explain after removing the low-level features we consider. This shows that speech-based models capture additional features that are important for early auditory cortex and understanding these further can help us better understand this part of the brain.
> > > > > >
> > > > > > I agree this is a potentially valuable contribution although it is not mentioned at all in the abstract. In fact, the abstract seems to emphasize the opposite conclusion:
> > > > > >
> > > > > > Using our direct approach, we find that both text-based and speech-based models align well with early sensory areas due to shared low-level features.
> > > > > >
> > > > > > This finding is not entirely new. For example, the Li et al. (2013) paper cited in their response shows something similar. If this is going to be a major focus of the paper it needs to be motivated and described as such and contrasted with the prior work in terms of what is new.
> > > > > >
> > > > > > The second contribution flagged is: Another surprising result is that, in contrast to the alignment of speech-based models in the early auditory cortex, the alignment in late language regions is almost entirely due to low-level features. The surprise is coming from the fact that the predictive performance of the late language regions is still good and almost on par with text-based models, and also from the fact that several recent works have proposed speech-based models for good models of language in the brain [Yuanning Li et al. 2023 Nature Neuroscience, Chen et al. 2023 Arxiv]. Our work clearly demonstrates the importance of careful consideration of the reasons behind the alignment that is observed between models and human brain recordings.
> > > > > >
> > > > > > The DNN audio models tested here only see 2-second stimuli and are audio-based, so the fact that including audio controls has a greater impact in high-level language regions strikes me as confirmatory. The Li paper is focused on characterizing responses to speech in the STG again mostly focused on short timescales, not high-level language regions in the STS and frontal cortex, like those canonically associated with high-level language regions. I think extending the window beyond 2 seconds would be valuable even if it doesn’t improve prediction accuracy, since that provides a stronger demonstration of the model’s failure and a stronger contrast with the text-based models.
> > > > > >
> > > > > > Even though the finding is not terribly surprising in my opinion, I still find it a useful contribution if framed appropriately, since it demonstrates a limitation of existing audio models that needs to be addressed in future work.

---

> > > > > > > ### Comment · Reviewer_cgRE · 2023-11-22
> > > > > > > **Comments on Methods**
> > > > > > >
> > > > > > > Downsampling is typically applied to evenly spaced samples. What is unclear to me is how you handle the fact that the words are not evenly spaced?
> > > > > > >
> > > > > > > I do not see an appendix section J.
> > > > > > >
> > > > > > > A proper noise ceiling estimates the variance explained in the absence of noise. It is not clear why the procedure they describe estimates this quantity. They have a small number of subjects and each subject has noisy data and so the predictions for the left-out subject are likely to be themselves highly imperfect. The fact that this was used in a prior paper does not mean that the procedure is appropriate. One solution would be to call it something else, e.g. “Cross-subject prediction accuracy” and then explain clearly in the text what the quantity is and how it differs from a noise ceiling.
> > > > > > >
> > > > > > > I remain highly skeptical about calculating a noise ceiling for visually presented text in A1 (and vice versa). I have a hard time believing that you have any reliable signal. Moreover, the voxel selection procedure you are using is almost certainly going to substantially bias the noise ceiling upwards since you are explicitly selecting voxels with a reliable value. What would happen if you selected voxels in one half of your dataset and then measured the noise ceiling in another set? What would the variance be? Could you even sensibly distinguish the noise ceiling from 0?
> > > > > > >
> > > > > > > I still don’t totally follow what this means: “To clarify, we first cross-validated on the training data set stimuli. Then for each fold in the banded ridge regression cross-validation, we trained the ridge regression model to remove low-level features to that particular fold. Overall, we performed a 10-fold cross-validation scheme as the training dataset stimuli consists of 10 different stories.” Are you saying that you learned the weights on the training set, selecting the regularization parameter via nested cross-validation on the training set, and then you applied the learned weights on test to derive a prediction and subtracted out the prediction?
> > > > > > >
> > > > > > > You do not answer this question: For the banded ridge regression, were the lambdas specified separately varied for each feature set? How do we know that this range is sufficient? It is highly sensitive to the scale of the features. How fine was the grid search? You repeat what was said in the methods and say it has been “thoroughly tested”.
> > > > > > >
> > > > > > > There are several places where the authors provide additional clarifications in their response to me, but do not indicate if and how the manuscript will be changed to improve clarity.

---

> > > > > > > > ### Author Response · Authors · 2023-11-22
> > > > > > > >
> > > > > > > > *Thank you for the detailed response and feedback! We are very thankful that we can get your feedback as an expert reviewer. Your suggestions have helped us strengthen our message and will improve the manuscript as we update it to fully reflect your feedback in the coming days.*
> > > > > > > >
> > > > > > > >
> > > > > > > > **Novelty of findings and framing:**
> > > > > > > >
> > > > > > > > Thanks for taking the time to clarify.
> > > > > > > > * We are happy to hear that you find the results we highlighted in our response valuable.
> > > > > > > > * We agree that we can further work on highlighting the most surprising findings in the text, and your feedback has been helpful in guiding us in that. We will update the abstract, introduction, and discussion to highlight the two results we discussed in the response.
> > > > > > > > * In the short discussion period, we gave priority to completing new analyses and addressing the concerns from all reviewers so we were not able to make all changes to the manuscript that we wanted yet. We will continue to update the manuscript in the coming days.
> > > > > > > >
> > > > > > > > **Downsampling is typically applied to evenly spaced samples. What is unclear to me is how you handle the fact that the words are not evenly spaced?**
> > > > > > > >
> > > > > > > > * We leverage improvements in downsampling techniques, which apply to non-evenly spaced word presentations, as is the case for our stimuli.
> > > > > > > > * To downsample the semantic representations, we use a 3-lobe Lanczos filter centered at the midpoint of each word presentation, which smoothly interpolates the value of a signal between its end points.
> > > > > > > > * Similar downsampling approaches have been leveraged by previous works which use non-evenly spaced stimuli [Jain & Huth 2018 NeurIPS, Jain & Huth 2020, NeurIPS, Antonello et al. 2021 NeurIPS, Oota et al. 2023 NeurIPS]
> > > > > > > >
> > > > > > > > *[Jain & Huth, 2018], Incorporating context into language encoding models for fmri. In NIPS-2018*
> > > > > > > >
> > > > > > > > *[Jain et al. 2020], Interpretable multi-timescale models for predicting fMRI responses to continuous natural speech, In NeurIPS-2020*
> > > > > > > >
> > > > > > > > *[Antonello et al. 2021], Low-Dimensional Structure in the Space of Language Representations is Reflected in Brain Responses, In NeurIPS-2021*
> > > > > > > >
> > > > > > > > *[Oota et al. 2023 NeurIPS] Joint processing of linguistic properties in brains and language models. NeurIPS-2023*
> > > > > > > >
> > > > > > > > **I do not see an appendix section J.**
> > > > > > > > * We apologize for this oversight. We had **Appendix section J** prepared but forgot to upload the final version to the system. We have updated the manuscript with the latest version now.
> > > > > > > >
> > > > > > > > **A proper noise ceiling estimates the variance explained in the absence of noise. It is not clear why the procedure they describe estimates this quantity. They have a small number of subjects and each subject has noisy data and so the predictions for the left-out subject are likely to be themselves highly imperfect. The fact that this was used in a prior paper does not mean that the procedure is appropriate. One solution would be to call it something else, e.g. “Cross-subject prediction accuracy” and then explain clearly in the text what the quantity is and how it differs from a noise ceiling.**
> > > > > > > >
> > > > > > > > * We agree with the reviewer that the procedure that we borrow from previous work to estimate the noise ceiling is imperfect. We will follow the reviewer’s suggestion and call this normalization factor the cross-subject prediction accuracy instead of noise ceiling and will add a discussion about the possible drawbacks.
> > > > > > > >
> > > > > > > > **I remain highly skeptical about calculating a noise ceiling for visually presented text in A1 (and vice versa). I have a hard time believing that you have any reliable signal. Moreover, the voxel selection procedure you are using is almost certainly going to substantially bias the noise ceiling upwards since you are explicitly selecting voxels with a reliable value. What would happen if you selected voxels in one half of your dataset and then measured the noise ceiling in another set? What would the variance be? Could you even sensibly distinguish the noise ceiling from 0?**
> > > > > > > >
> > > > > > > > * The reviewer is correct that the cross-subject prediction accuracy in A1 during reading and V1 during listening is low.
> > > > > > > > * This is in fact what we show in **Fig 20 in Appendix section I** which we added in response to the reviewer’s concern.
> > > > > > > > * The point we wanted to make in the previous response was that because this normalization factor is indeed small in these regions during the non-presentation modality, one should be skeptical of large normalized predictivity in these cases which may be the result of this low normalization factor.
> > > > > > > > * However, we show that the final normalized prediction accuracy in the sensory regions of the non-presentation modality is still in fact very low (Fig 1: checkered bars in early visual, and solid bars in early auditory), so the low cross-subject prediction accuracy in these regions does not change our conclusions.
> > > > > > > > * Following the reviewer’s suggestion to present the cross-subject prediction accuracy along with the normalized predictivity will ensure that the normalized predictivity is interpreted appropriately.

---

> > > > > > > > > ### Author Response · Authors · 2023-11-22
> > > > > > > > >
> > > > > > > > > **I still don’t totally follow what this means: “To clarify, we first cross-validated on the training data set stimuli. Then for each fold in the banded ridge regression cross-validation, we trained the ridge regression model to remove low-level features to that particular fold. Overall, we performed a 10-fold cross-validation scheme as the training dataset stimuli consists of 10 different stories.” Are you saying that you learned the weights on the training set, selecting the regularization parameter via nested cross-validation on the training set, and then you applied the learned weights on test to derive a prediction and subtracted out the prediction?**
> > > > > > > > >
> > > > > > > > >
> > > > > > > > > Yes, precisely. To be sure we’re on the same page, let us break down the procedure using the following notation:
> > > > > > > > > * Low-level features:  $L_{train}$ and $L_{test}$, corresponding to training and test data for one cross validation fold
> > > > > > > > > * Model representations: $M_{train}$ and $M_{test}$
> > > > > > > > > * Brain recordings: $B_{train}$ and $B_{test}$
> > > > > > > > >
> > > > > > > > > Within each cross validation fold of the banded ridge, we:
> > > > > > > > > 1) train a ridge regression model $L_{train}$ -> $M_{train}$ for the removal of a specific low-level feature (the best ridge parameter is obtained via nested cross validation) and obtain the weights of the corresponding best ridge model,
> > > > > > > > > 2) using these weights and $L_{train}$ and $L_{test}$, we make predictions for $M_{train}$ and $M_{test}$ ($M_{train}^{hat}$, $M_{test}^{hat}$),
> > > > > > > > > 3) obtain the training residuals $M_{train}^{res}$ by subtracting $M_{train}^{hat}$ from $M_{train}$, and the test residuals $M_{test}^{res}$ by subtracting the $M_{test}^{hat}$ from $M_{test}$,
> > > > > > > > > 4) and finally train the parameters of the outer banded ridge fold $M_{train}^{res}$ -> $B_{train}$ using the train residual model representations and the train brain data, and
> > > > > > > > > 5) test the banded ridge by applying these learned weights to $M_{test}^{res}$ and comparing these predictions via Pearson correlation with $B_{test}$.
> > > > > > > > >
> > > > > > > > > We will clarify this procedure in the text.
> > > > > > > > >
> > > > > > > > > **You do not answer this question: For the banded ridge regression, were the lambdas specified separately varied for each feature set? How do we know that this range is sufficient? It is highly sensitive to the scale of the features. How fine was the grid search? You repeat what was said in the methods and say it has been “thoroughly tested”.**
> > > > > > > > >
> > > > > > > > > * We optimized the lambda parameters for each feature separately within the range of 10 to 10$^3$, spaced logarithmically.
> > > > > > > > > * The best lambda parameter was chosen using the validation data in each cross-validation fold.
> > > > > > > > > * Our selection of a range of lambda parameters was based on previous studies using the same moth-radio-hour dataset [Jain et al. 2018 NIPS, Deniz et al. 2019 Journal of Neuroscience, Jain et. al. 2020 NeurIPS, Oota et al. 2023 NeurIPS].
> > > > > > > > >
> > > > > > > > > *[Jain & Huth, 2018], Incorporating context into language encoding models for fmri. In NIPS-2018*
> > > > > > > > >
> > > > > > > > > *[Deniz et al. 2019], The representation of semantic information across human cerebral cortex during listening versus reading is invariant to stimulus modality. Journal of Neuroscience. 2019*
> > > > > > > > >
> > > > > > > > > *[Jain et al. 2020], Interpretable multi-timescale models for predicting fMRI responses to continuous natural speech, In NeurIPS-2020*
> > > > > > > > >
> > > > > > > > > *[Oota et al. 2023 NeurIPS] Joint processing of linguistic properties in brains and language models. NeurIPS-2023*
> > > > > > > > >
> > > > > > > > > **There are several places where the authors provide additional clarifications in their response to me, but do not indicate if and how the manuscript will be changed to improve clarity.**
> > > > > > > > >
> > > > > > > > > * We value your feedback and will incorporate all clarifications that we provided to you and the remaining reviewers. We worked hard to update the manuscript with new analyses in the short time that we had during the discussion period, and will focus on incorporating the remaining clarifications in the coming days.

---

> > > > > > > > > > ### Comment · Reviewer_cgRE · 2023-12-02
> > > > > > > > > > **Summary thoughts**
> > > > > > > > > >
> > > > > > > > > > The authors have in general been very responsive to the reviewer feedback, which I appreciate.
> > > > > > > > > >
> > > > > > > > > > The authors have addressed a bunch of my methodological concerns. I'm still not convinced that the cross-brain prediction score has any validity for early sensory areas for the non-preferred modality, i.e., I'm not convinced they can differentiate the score reliably from 0. The above 0 predictions I think could entirely be due to a circular voxel selection criteria. I get the author's point that the bias will be to deflate the corrected values, but I'm still uncomfortable with a main text figure whose values are wildly off. I am going to increase my soundness score from 2 to 3.
> > > > > > > > > >
> > > > > > > > > > In my opinion, the author's main contribution is to show that speech models like wav2vec2.0 and Whisper don't capture high-level language information in high-level language regions and that much of their predictivity can be accounted for by lower-level acoustic/speech features. I think this is a useful contribution, though not particularly groundbreaking or surprising, and does highlight an important gap in speech modeling. The main data point supporting this is that the performance of speech models declines by 2/3 when you add low-level acoustic/speech features. There is still definitely some non-trivial predictivity, and it's hard to know what that reflects. Plus they are predicting from 2 seconds of audio, so it doesn't seem that surprising that there is a lack of high-level language info.
> > > > > > > > > >
> > > > > > > > > > To me, this paper is close but not quite there. I am raising my score to a 5. I think if they nailed one core claim, it would be a 6 for me, but I'm not completely convinced.

---

### Official Review · Reviewer_1bg5 · 2023-11-09

**Soundness:** 1 poor
**Presentation:** 2 fair
**Contribution:** 1 poor
**Rating:** 3
**Confidence:** 4

**Summary:**

A research paper submitted by the authors raises concerns about the quality and validity of the study. The reviewer argues that the study is questionable because it is based on fMRI recordings obtained from only six subjects, which is too small a sample size for any publication. Moreover, the methodology used in the study is not new, and the results are obvious, merely showing that LLMs are predictable to the human brain.

While the reviewer acknowledges that they may have missed something, they emphasize that the study lacks methodological novelty and scientific rigor. In other words, the research does not bring anything new to the field and fails to meet the basic standards of scientific research.

Overall, the reviewer's critique suggests that the authors need to revisit their study and address the concerns raised by the reviewer in order to produce a more compelling and scientifically robust piece of research.

**Strengths:**

It is difficult to determine the strength of the paper, as the contribution appears weak. Furthermore, the use of a small dataset downloaded from the internet, combined with existing analysis methods, fails to provide any novelty. The authors have not even attempted to persuade the reader that such a study makes sense. It is evident that language modeling algorithms (LLMs) are trained to replicate human language. Therefore, it is likely that human brain activity would follow language that sounds or looks natural, just as it would if delivered by a human.

**Weaknesses:**

Lack of methodological novelty; small dataset from another study without validation of its relevance for the authors' analysis; and, most importantly, a lack of argumentation justifying the study.

**Questions:**

1. Why is such a small fMRI dataset used?
2. What are the technical, methodological, and scientific contributions that would interest the ICLR audience?

---

> ### Author Response · Authors · 2023-11-22
>
> *We thank the reviewer for their valuable comments and suggestions. We address the reviewer's concerns point by point below. We hope that the reviewer can reevaluate our work in light of these clarifications.*
>
> **Size of fMRI dataset**
>
> We thank the reviewer for this suggestion.
>
> * The fMRI dataset we use is a well-known public dataset that has previously been used in many publications in both ML and Neuroscience venues (ML: [Jain & Huth 2018 NeurIPS, Jain et al. 2020 NeurIPS, Antonello et al. 2021 NeurIPS, Lamarre et al. 2022 EMNLP, Vaidya et al. 2022 ICML;], Neuroscience: [Huth et al. 2016 Nature, Huth et al. 2017 Journal of Neuroscience, Deniz et al. 2019 Journal of Neuroscience]).
> * For the kind of analyses that we do and that are common in this area of research, the number of samples per participant is more important than the number of subjects because the predictive models are trained independently for each participant. So, having more samples per participant helps us learn a better predictive model.
> * This dataset is one of the largest datasets in terms of samples per participant (~4000 samples), which is the main reason for its frequent use.
> * Our results also clearly show that this dataset is sufficient to learn a good predictive model, as we show that we can predict up to 75% of the explainable variance for held-out brain recordings that were not used for training (e.g., Fig. 3).
>
> *[Huth et al. 2016], Natural speech reveals the semantic maps that tile human cerebral cortex. In Nature Neuroscience, 2016*
>
> *[De Heer et al. 2017], The hierarchical cortical organization of human speech processing, Journal of Neuroscience, 2017*
>
> *[Jain & Huth, 2018], Incorporating context into language encoding models for fmri. In NIPS-2018*
>
> *[Deniz et al. 2019], The representation of semantic information across human cerebral cortex during listening versus reading is invariant to stimulus modality. Journal of Neuroscience, 2019*
>
> *[Jain et al. 2020], Interpretable multi-timescale models for predicting fMRI responses to continuous natural speech, In NeurIPS-2020*
>
> *[Antonello et al. 2021], Low-Dimensional Structure in the Space of Language Representations is Reflected in Brain Responses, In NeurIPS-2021*
>
> *[Lamarre et al. 2022] Attention weights accurately predict language representations in the brain, EMNLP-2022*
>
> *[Vaidya et al. 2022] Self-supervised models of audio effectively explain human cortical responses to speech, ICML-2022*
>
> **Contributions and implications of our work**
>
> Our main contributions are the fine-grained analysis approach of the contributions of different types of information to the brain alignment of speech-based and text-based language models, and the scientific findings of this analysis. We summarize the implications of our findings below and include these in the updated manuscript (**see Appendix section H**).
>
> * Our findings have direct implications for both machine learning and cognitive neuroscience.
> * First, we show that even during speech-evoked brain activity (i.e., listening), the alignment of speech-based models trails behind that of text-based models in the late language regions. More importantly, our results demonstrate that the alignment of speech-based models with these regions is almost entirely explained by low-level stimulus features. Since these regions are purported to represent semantic information, this finding implies that contemporary speech-based models lack brain-relevant semantics. This suggests that new machine learning approaches are needed to improve speech-based models. Furthermore, these results imply that observed similarities between speech-based models and brain recordings in the past [Vaidya et al. 2022 ICML, Millet et al. 2022 NeurIPS] are largely due to low-level information and not semantics, which is important to take into account when interpreting the similarity between language representations in speech-based models and the brain.
>
> * Second, we observe that phonological features explain the most variance during listening for speech-based models, whereas ``number of letters" explains the most variance during reading for text-based models. This result offers us a glimpse into the mechanisms underlying language processing in the brain.
>
> * Third, we demonstrate a direct residual approach to identify the contribution of specific features to brain alignment. To our knowledge, there is no better alternative to selectively remove information from language models to probe their impact on brain alignment. Using this approach, it is possible to investigate how the human brain processes language during both reading and listening at a finer scale than before.
>
> *[Vaidya et al. 2022] Self-supervised models of audio effectively explain human cortical responses to speech, ICML-2022*
>
> *[Millet et al. 2022] Toward a realistic model of speech processing in the brain with self-supervised learning. NeurIPS-2022*

---

> > ### Comment · Reviewer_1bg5 · 2023-11-22
> > **No change in decision**
> >
> > The reviewer appreciates the author's feedback. However, despite the long recordings and machine learning training settings that have been optimized for the small subject sample, the results must be validated on a larger participant sample. The current dataset may be popular, but it is important to ensure the validity of the findings. The authors should also provide a more robust defense of the novelty of their approach, and clarify their contribution to the field of cognitive neuroscience.

---

### Official Review · Reviewer_5at5 · 2023-11-10

**Soundness:** 3 good
**Presentation:** 4 excellent
**Contribution:** 3 good
**Rating:** 6
**Confidence:** 3

**Summary:**

The researchers are trying to understand what kind of information that text-based and speech-based language models are actually predicting about brain activity.  They are using a controlled experimental setup with a known fMRI dataset to systematically investigate brain alignment of language models by eliminating specific low-level textual, speech, and visual features from model representations. Finding reveal that both text and speech-based models align with the brain's early sensory areas due to common low-level features. But when these features were removed, text-based models still aligned well with brain regions involved in language processing, while speech-based models did not. This was unexpected and suggests that speech-based models might need improvement to better mimic brain-like language processing.

**Strengths:**

The findings of this research are important in computational linguistics and neuroscience.

The paper provides insights into the potential of text-based language models to capture deep linguistic semantics by showing the models can predict brain activity in language-processing regions without low-level features. The observation regarding speech-based models suggests a possible direction to further improve their capability.

The controlled experimental design and statistical approach seem rigorous to me. The clarity of the paper is good and easy to follow.

**Weaknesses:**

I’m not familiar with the six-participate dataset and not certain if the limited scope of the datasets could affect the generalizability of the results. It would be nice if the author could discuss how to translate the findings translate to different languages, models, and datasets.

While the author describes details regarding their experiment setup, it would benefit the community if the author could publish their implementation, especially the low-level feature removal and data preprocessing. These aspects are critical for ensuring reproducibility and are not entirely clear to me. Making this information available would significantly enhance the paper's utility and impact.

**Questions:**

I'm seeking to better understand the application of ridge regression in the context of your study. Specifically, when you remove low-level feature vectors from pre-trained features, could this process potentially alter or diminish the representation of higher-level features? In other words, might the removal of these low-level signals inadvertently affect the model's ability to process more complex, abstract linguistic features that are also captured in these representations?

I would be interested to see more discussion for Figure 10.

---

> ### Author Response · Authors · 2023-11-22
>
> *We thank the reviewer for their encouraging words. We address the reviewer's concerns point by point below.*
>
> **I’m not familiar with the six-participate dataset and not certain if the limited scope of the datasets could affect the generalizability of the results. It would be nice if the author could discuss how to translate the findings translate to different languages, models, and datasets.**
>
> We thank the reviewer for this suggestion.
> * The fMRI dataset we use is a well-known public dataset that has previously been used in many publications in both ML and Neuroscience venues (ML: [Jain & Huth 2018 NeurIPS, Jain et al. 2020 NeurIPS, Antonello et al. 2021 NeurIPS, Lamarre et al. 2022 EMNLP, Vaidya et al. 2022 ICML;], Neuroscience: [Huth et al. 2016 Nature, Huth et al. 2017 Journal of Neuroscience, Deniz et al. 2019 Journal of Neuroscience]).
> * For the kind of analyses that we do and that are common in this area of research, the number of samples per participant is more important than the number of subjects because the predictive models are trained independently for each participant. So, having more samples per participant helps us learn a better predictive model.
> * This dataset is one of the largest datasets in terms of samples per participant (~4000 samples), which is the main reason for its frequent use.
> * Our results also clearly show that this dataset is sufficient to learn a good predictive model, as we show that we can predict up to 75% of the explainable variance for held-out brain recordings that were not used for training (e.g., Fig. 3).
>
> We expect the main results of the current study to persist even if a similar experiment and analysis were to be conducted in a different language, model, or a dataset. There is, however, value in actually testing this hypothesis and confirming this prediction. We hope to see future work replicating our findings across different languages and datasets. To our knowledge, there are very few naturalistic datasets in different languages. Therefore, to be able to test this prediction it is necessary to collect neuroimaging data using different languages.
>
> *[Huth et al. 2016], Natural speech reveals the semantic maps that tile human cerebral cortex. In Nature Neuroscience, 2016*
>
> *[De Heer et al. 2017], The hierarchical cortical organization of human speech processing, Journal of Neuroscience, 2017*
>
> *[Jain & Huth, 2018], Incorporating context into language encoding models for fmri. In NIPS-2018*
>
> *[Deniz et al. 2019], The representation of semantic information across human cerebral cortex during listening versus reading is invariant to stimulus modality. Journal of Neuroscience, 2019*
>
> *[Jain et al. 2020], Interpretable multi-timescale models for predicting fMRI responses to continuous natural speech, In NeurIPS-2020*
>
> *[Antonello et al. 2021], Low-Dimensional Structure in the Space of Language Representations is Reflected in Brain Responses, In NeurIPS-2021*
>
> *[Lamarre et al. 2022] Attention weights accurately predict language representations in the brain, EMNLP-2022*
>
> *[Vaidya et al. 2022] Self-supervised models of audio effectively explain human cortical responses to speech, ICML-2022*
>
> **While the author describes details regarding their experiment setup, it would benefit the community if the author could publish their implementation, especially the low-level feature removal and data preprocessing. These aspects are critical for ensuring reproducibility and are not entirely clear to me. Making this information available would significantly enhance the paper's utility and impact.**
>
> We thank the reviewer for pointing out open science practices, which are very important to us. We agree with the reviewer that it is critical to make our code openly available so that others can reproduce our results and easily use our approach for their own research questions. Therefore, we are going to publish our implementation, including how we remove low-level features. In this study, we used an openly available fMRI dataset which includes preprocessed BOLD responses. The details of data preprocessing can be found in the original article [Deniz et al. 2019 Journal of Neuroscience].
>
> *[Deniz et al. 2019], The representation of semantic information across human cerebral cortex during listening versus reading is invariant to stimulus modality. Journal of Neuroscience, 2019*

---

> > ### Author Response · Authors · 2023-11-22
> >
> > **I'm seeking to better understand the application of ridge regression in the context of your study. Specifically, when you remove low-level feature vectors from pre-trained features, could this process potentially alter or diminish the representation of higher-level features? In other words, might the removal of these low-level signals inadvertently affect the model's ability to process more complex, abstract linguistic features that are also captured in these representations?**
> >
> > We thank the reviewer for this insightful question.
> >
> > * Removal of low-level features can potentially have an impact on high-level, semantic representations due to possible correlations across the two types of features.
> > * This is true for our method as well as all other approaches employing encoding models.
> > * However, we want to point out that there is much high-level semantic information that is not impacted by the removal of low-level features because we show that late language regions are still predicted extremely well by text-based models even after removing the low-level features.
> > * Therefore the huge impact of removing low-level features from speech-based models on the prediction of late language regions cannot be due to confounding correlations between low-level features and high-level semantics. This is an important point and we will add a discussion of it in the paper.
> > * Overall, while there may be potential confounds between low- and high-level features, we do not believe such confounds affect the main conclusions of this particular study.
> >
> > **I would be interested to see more discussion for Figure 10.**
> >
> > Please find below the updated discussion about Fig. 10. The corresponding section B in the Appendix is also adjusted accordingly in the revised paper.
> >
> > * In Fig. 10, we report the whole brain alignment of each model normalized by the noise ceiling for the naturalistic reading and listening datasets.
> > * We show the average normalized brain alignment across subjects, layers, and voxels. Note that we are only averaging across voxels which have a statistically significant brain alignment.
> > * We perform the Wilcoxon signed-rank test to test whether the differences between text- and speech-based language models are  statistically significant.
> > * We found that all text-based models predict brain responses significantly better than all speech-based models in both modalities.
> > * Across text-based models, the whole brain alignment gradually diminishes when going from BERT to GPT-2 to FLAN-T5 both during reading and listening.
> > * Across speech-based models, stimulus modality had a significant effect. While Wav2Vec2.0 is the better performer during reading, Whisper aligns better with the whole brain during listening. The fact that Whisper is trained on a larger amount of speech data could be the reason underlying its better alignment during listening.

---

### Author Response · Authors · 2023-11-23
**Summary of our responses and revision:**

*We are grateful to all reviewers for their time and their constructive suggestions, which will further strengthen the impact of our work.*

Here we summarize the new analyses and results that we have included in the updated manuscript in response to the reviewers:

**Model-specific results vs averages across model type:**
- We now present the results for each individual speech-based language model (Wav2Vec2.0 and Whisper) for predicting A1 and late language regions during listening in the updated Appendix F (see Figure 17). Removal of low-level textual and speech features results in a significant performance decline in both models. However, the brain alignment of Wav2Vec2.0 is affected slightly more from this removal in A1.

**Layer-wise results**

- We want to clarify that each layer’s activations were used independently to build voxel-wise encoding models, and the results presented in the paper depict the average normalized brain alignment across all of the layers for both text-based and speech-based language models and their corresponding residuals. We now also present the layer-wise normalized brain alignment for the Wav2Vec2.0 model during listening, both before and after removal of one important low-level speech property: phonological features, as shown in Figure 18. (see in the updated Appendix G). Using the averaged results, we had shown that low-level features impact the ability to predict both A1 and late language regions. This layer-wise analysis clearly demonstrates that this key finding holds across individual layers.

**A1 results**

- For the early auditory analysis, we were asked to report the normalized brain alignment results separately for A1 during listening for both text- and speech-based language models, along with their residual performance after eliminating low-level stimulus features, which we now do in the updated Appendix J (see Figure 21). Our results show that speech-based models align better than text-based models during listening in A1. Removal of low-level textual features results in a similar performance drop for both types of models, whereas removal of low-level speech features results in a larger performance drop for speech-based compared to text-based language models.

**Baseline results with low level features**

- We now present the average normalized brain alignment for low-level stimulus features (textual, speech, and visual) during both reading and listening in the early sensory areas (early visual and auditory) in Figure 19 (see section G in Appendix). These results confirm our finding that low-level stimulus features are important for prediction of the early sensory processing regions, and that specifically speech models still explain significantly more variance in the early auditory cortex than the low-level features.

**Estimated noise-ceiling across ROIs (which we will now call cross-subject prediction accuracy as suggested by reviewer cgRE):**

- One potential concern is that because the noise ceiling in the sensory regions of the non-presentation modality is low (A1 during reading, and V1 during listening), one may expect that using this low noise ceiling to normalize the prediction performance of models may result in an overly inflated number. However, we see that the normalized predictivity in the sensory regions of the non-presentation modality is in fact quite low (Fig 1: checkered bars in early visual, and solid bars in early auditory). We also present the average noise ceiling estimate across subjects for both the reading and listening conditions  in the Appendix section I (see Figure 20).

**Contributions and implications of our work**
- We summarize the implications of our findings and include these in the updated manuscript (see Appendix section H).

We believe that these additional analyses and clarifications have significantly strengthened our work and helped us present the contributions in a way that best showcases their timeliness.

---

### Meta-Review · Area_Chair_94oW · 2023-12-04

**Metareview:**

In this paper the authors carry out an analysis to investigate the brain fMRI responses to activations of speech and text based large scale deep neural network models in the machine learning community.  The authors design the analysis by carefully removing low-level text, audio and visual features to observe their impact to the alignments with fMRI brain recordings.  They find that
both text and speech based large scale DNN models align well with early sensory areas due to shared low-level features. However, when those low level features are removed, text-based models continue to align well with later language regions while speech-based models fail to do that.  Overall, this is an interesting work and may have its value to the cognitive neuroscience community.  There are numerous concerns raised by the reviewers and the authors have put up a meticulous rebuttal to address them.  There has been a pretty constructive discussion on the work during the rebuttal period.  In the end, after going through rebuttal and the paper,  I tend to agree that there are some standing concerns. (For instance, what observation would be if the analysis window of speech is longer than 2 seconds to capture more linguistic information?)  Reviewers pointed out related work or approaches from other labs and literature.  I think it would be helpful to address them to make the paper stronger.  In general, the paper has its value and potential but it also needs improvements to make the case stronger and claims more convincing.  I would suggest the authors carefully consider some of the comments to improve the work and submit to a future venue.

**Justification For Why Not Higher Score:**

There are remaining concerns on the experimental design which I consider significant.  The paper has its value but can not be accepted given its current form.

**Justification For Why Not Lower Score:**

N/A

---

### Decision · Program_Chairs · 2024-01-16

Reject